# Collective production of hydrogen sulfide gas enables budding yeast lacking *MET17* to overcome their metabolic defect

**Sonal**[1]*, **Alex E. Yuan**[2], **Xueqin Yang**[3], **Wenying Shou**[1]*

**1** Centre for Life's Origins and Evolution, Department of Genetics, Evolution and Environment, University College London, London, United Kingdom, **2** University of Washington, Seattle, Washington, United States of America, **3** School of Environmental Science and Engineering, Sun Yat-sen University, Guangzhou, China

* sonalnd@gmail.com (S); wenying.shou@gmail.com (WS)

## Abstract

Assimilation of sulfur is vital to all organisms. In *S. cerevisiae*, inorganic sulfate is first reduced to sulfide, which is then affixed to an organic carbon backbone by the Met17 enzyme. The resulting homocysteine can then be converted to all other essential organosulfurs such as methionine, cysteine, and glutathione. This pathway has been known for nearly half a century, and *met17* mutants have long been classified as organosulfur auxotrophs, which are unable to grow on sulfate as their sole sulfur source. Surprisingly, we found that *met17Δ* could grow on sulfate, albeit only at sufficiently high cell densities. We show that the accumulation of hydrogen sulfide gas underpins this density-dependent growth of *met17Δ* on sulfate and that the locus *YLL058W* (*HSU1*) enables *met17Δ* cells to assimilate hydrogen sulfide. Hsu1 protein is induced during sulfur starvation and under exposure to high sulfide concentrations in wild-type cells, and the gene has a pleiotropic role in sulfur assimilation. In a mathematical model, the low efficiency of sulfide assimilation in *met17Δ* can explain the observed density-dependent growth of *met17Δ* on sulfate. Thus, having uncovered and explained the paradoxical growth of a commonly used "auxotroph," our findings may impact the design of future studies in yeast genetics, metabolism, and volatile-mediated microbial interactions.

## Introduction

Sulfur metabolism is vital to all organisms and produces a range of essential metabolites. Perhaps the best-known organosulfurs—sulfur-containing organic compounds—are the essential amino acids methionine and cysteine. The relevance of sulfur metabolites, however, extends far beyond protein synthesis. AdoMet, an activated form of methionine, serves as a universal methyl donor for nucleotides, proteins, and small metabolites, and perturbations of AdoMet metabolism are implicated in liver pathologies [1]. Glutathione, generated from cysteine, is crucial for cellular redox homeostasis [2]; decreased glutathione levels lead to oxidative stress, which has been implicated in ageing and neurodegeneration [3,4]. Hydrogen sulfide (H₂S), an inorganic gas that can be produced during organosulfur metabolism, has therapeutic potential

model are available at: doi.org/10.5281/zenodo.
10142030.

**Funding:** This project has received funding from
the European Union's Horizon 2020 research and
innovation programme under the Marie
Skłodowska-Curie grant agreement no.
101025821. The US National Science Foundation
(award number 1917258) funded S and WS. WS
was additionally funded by a Royal Society
Wolfson Foundation Fellowship and a
professorship from the Academy of Medical
Sciences (AMS) and the Government Department
of Business, Energy and Industrial Strategy (BEIS).
AEY was also funded by AMS. XY was funded by
the China Scholarship Council (202006380122).
The funders had no role in study design, data
collection and analysis, decision to publish, or
preparation of the manuscript.

**Competing interests:** The authors have declared
that no competing interests exist.

**Abbreviations:** DMS, dimethyl sulfide; H$_2$S,
hydrogen sulfide; *HSU1*, *Hydrogen Sulfide
Utilizing-1*; OAH, *O*-acetyl homoserine; OAS, *O*-
acetyl serine; SMM, S-methylmethionine.

in gastrointestinal, cardiovascular, inflammatory, and nervous systems [5,6]. Thus, a deeper understanding of sulfur metabolism could facilitate wide-ranging biomedical advances.

Besides these critical roles in cell physiology, secreted sulfur metabolites can also influence population-level behavior in microbes. For example, in the budding yeast *Saccharomyces cerevisiae*, populations can expand the range of temperatures they can tolerate by leveraging the antioxidant properties of secreted glutathione [7]. As another example, H$_2$S is thought to promote population synchrony during ultradian respiratory oscillations in aerobic continuous cultures by inhibiting the respiratory chain, and exogenous sulfide can shift the phase of these oscillations [8,9]. In fact, microorganisms release a constellation of volatile sulfur compounds [10], which have been of interest to the food industry (e.g., in optimizing wine aroma [11]), but the significance of many of these compounds to the microbes themselves is poorly understood.

Unlike humans, yeast can synthesize essential organosulfurs de novo by assimilating inorganic sulfates (SO$_4^{2-}$) from their environment. Various genes of sulfur metabolism were discovered in the budding yeast through genetic screens for organosulfur auxotrophs—mutants that can only grow when an organosulfur is supplemented and are otherwise unable to grow on sulfate [12]. These studies, together with biochemical analyses, led to the current model of sulfur metabolism in yeast [12]. In a simplified version of this model (Fig 1A), sulfate is reduced to hydrogen sulfide (H$_2$S) through a series of enzymatic reactions. Hydrogen sulfide then reacts with *O*-acetyl homoserine (OAH; Fig 1A) to form homocysteine, an organosulfur that can be interconverted to other organosulfurs (Fig 1A, purple). Some organosulfurs (e.g., cysteine) can be broken down to release H$_2$S and sulfite through dedicated pathways.

The *MET17* gene, also known as *MET15* or *MET25* [13–15], catalyzes homocysteine synthesis by reacting H$_2$S with OAH (i.e., displaying OAH sulfhydrylase activity) [16–18]. Although purified Met17 protein could also catalyze cysteine synthesis by reacting H$_2$S with *O*-acetyl serine (OAS) (i.e., displaying OAS sulfhydrylase activity [19]), this activity was low [18] and unlikely to be relevant in vivo in *S. cerevisiae* [20]. Other yeast species such as *K. lactis*, *Y. lipolytica*, and *S. pombe* have dedicated enzymes to catalyze these 2 reactions [21]. However, alternate enzymes with OAS sulfhydrylase activity have thus far not been characterized in *S. cerevisiae*. Thus, Met17 was assigned as the sole enzyme for synthesizing homocysteine—the precursor to all other forms of organosulfurs (Fig 1A). *met17* mutants are considered organosulfur auxotrophs, and the *met17* deletion mutation (*met17Δ*) [22] is commonly used in genetic studies of yeast, for example, as a background mutation in the yeast deletion library [23,24].

Surprisingly, we found that *met17Δ* yeast can, in fact, grow on sulfate without any organosulfur supplements, albeit in a density-dependent manner. Here, we show that density dependence is mediated via the volatile metabolite H$_2$S, which can be assimilated into organosulfurs through the *YLL058W* (*HSU1*) gene in the absence of *MET17*. A mathematical model based on low activity of Hsu1 can explain the density-dependent growth of *met17Δ* on sulfate. We conclude by reconciling conflicts in earlier studies, highlighting considerations for studying gas-mediated microbial interactions, and speculating on the functions of *HSU1* in wild-type yeast.

## Results

### *met17Δ* yeast show density-dependent growth on sulfate

*met17* mutants were first identified in genetic screens for organosulfur auxotrophs [25] and have long been thought to be unable to grow on inorganic sulfate [12]. On synthetic minimal medium (SD) containing sulfate as the sole sulfur source, the wild-type prototrophic strain

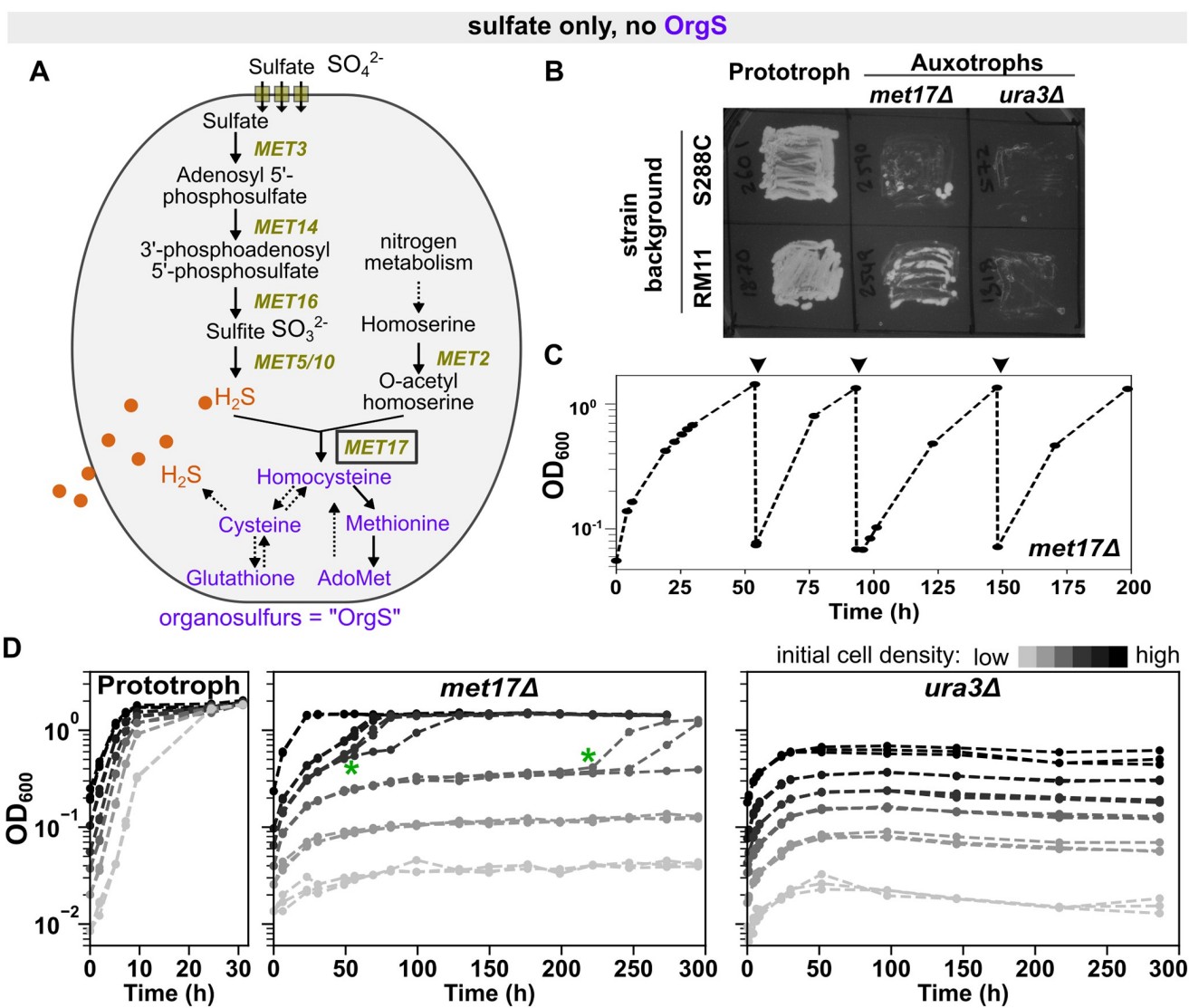

**Fig 1. *met17Δ* yeast show density-dependent growth on sulfate.** **(A)** Schematic summarizing sulfate assimilation in *S. cerevisiae*. Sulfate $SO_4^{2-}$ is reduced to sulfide $S^{2-}$ (orange) through a series of enzymatic reactions, and Met17 then combines sulfide with a nitrogenous compound (*O*-acetyl homoserine) to generate homocysteine. All organosulfurs (purple) are interconvertible via the transsulfuration pathway (cysteine, glutathione) and the methyl cycle (methionine, AdoMet). Solid and dotted lines represent one-step and multistep reactions, respectively. Note that according to this conventional understanding, *met17* mutants should not be able to generate organosulfurs from sulfate and, therefore, should require supplementation of organosulfur in their growth media. **(B)** Auxotrophy of *met17Δ* is leaky in both S288C and RM11 backgrounds. On agar plates containing synthetic minimal medium (SD: with sulfate, but without organosulfurs), patches of prototrophic yeast grew a dense lawn and patches of the uracil auxotroph *ura3Δ* remained clear, while patches of *met17Δ* showed papillae or patchy growth. Plate was imaged 3 days after patching. Prototrophic strains were WY2601 and WY1870, *met17Δ* were WY2590 and WY2548, and *ura3Δ* were WY572 and WY1318. **(C)** *met17Δ* yeast can be repeatedly passaged on sulfate. A liquid culture of *met17Δ* (WY2035) in SD medium could persistently grow to saturation upon repeated dilutions (arrowheads). **(D)** *met17Δ* show density-dependent growth on sulfate. Darker shades of gray represent higher initial cell densities. At each starting density, 3 identical liquid cultures were set up. For *met17Δ* (middle panel; WY2035), all 3 cultures at higher initial cell densities grew deterministically, while stochasticity in lag time was observed at intermediate cell densities (green asterisks) and the lowest cell densities did not grow. In contrast, prototrophs (left panel; WY1870) grew to saturation from all initial densities, while *ura3Δ* (right panel; WY1318) showed only residual growth at any density when uracil was withheld. In (**C**) and (**D**), strains were of RM11 background and population growth was recorded as optical density at 600 nm ($OD_{600}$). Each trendline represents a 7-ml culture in an 18-mm diameter glass tube with a loosely fitted plastic cap. WY2035 require lysine supplementation in SD medium, which was maintained throughout. The data underlying this figure can be found in S1 Data.

grew a dense lawn as expected, while the patch of a *ura3* deletant (*ura3Δ*, auxotrophic for uracil) remained clear (Fig 1B). Surprisingly, in the patches of the *met17* deletion (*met17Δ*) mutant, individual colonies or patchy growth would appear after 1–3 days of incubation (Fig 1B). Furthermore, *met17Δ* yeast could sometimes grow to high cell densities in liquid SD medium over repeated passaging (Fig 1C). This indicated that we were not merely observing residual growth fueled by organosulfurs that the cells had accumulated during exponential growth in methionine-supplemented medium. Similar observations have since been reported by other labs [26,27], and chromatographic analyses have confirmed that organosulfur contaminants occur in negligible quantities in the commercially available yeast nitrogenous base that was used to prepare SD medium [27].

The seemingly erratic growth behavior of *met17Δ* on sulfate could potentially be explained by growth being dependent on the initial cell density of liquid cultures (Fig 1D, middle panel). At high initial cell densities, the growth of *met17Δ* on sulfate was deterministic. At very low initial cell densities, the cultures did not grow. Strikingly, at intermediate cell densities, growth in the cultures was stochastic: Three replicate cultures started from the same parent culture would deviate in the lag time before population growth took off (Fig 1D, green asterisks in middle panel). As a comparison, prototrophic wild-type yeast immediately started to grow at all initial cell densities and reached saturation (Fig 1D, left panel), whereas uracil auxotrophs only showed residual growth at any cell density in the absence of uracil (Fig 1D, right panel; all cultures showed a fixed-fold increase in turbidity without reaching saturation). Note that all genotypes were growing exponentially in minimal medium (SD) supplemented to compensate for auxotrophies, before being washed and transferred to media wherein organosulfurs were withheld for *met17Δ* and uracil was withheld for *ura3Δ*. The growth dynamics of *met17Δ* indicate that these deletants are neither true auxotrophs nor true prototrophs but can overcome their auxotrophy at sufficiently high cell densities.

While our initial observations (Fig 1C and 1D) were made in yeast of the background RM11, we also observed a similar phenomenon in the S288C background (Fig A in S1 Figs). The phenotype was, however, weaker: S288C *met17Δ* grew to high turbidity in SD medium, but growth did not persist after dilution (Fig Ai in S1 Figs) and required considerably higher initial cell densities as compared to RM11 (Fig Aii in S1 Figs).

## Density-dependent growth of *met17Δ* on sulfate is mediated by hydrogen sulfide

Density-dependent growth in microbial populations is often mediated by the release of a chemical, which, upon reaching a critical concentration, enables cells to grow and divide. Because *met17Δ* are perturbed at the enzymatic assimilation of the volatile compound hydrogen sulfide ($H_2S$), we asked whether the growth mediator of *met17Δ* might be volatile. To test this, we limited gas exchange by using parafilm to seal the loose-fitting plastic lids on the culture tubes and asked if the growth outcomes of *met17Δ* populations at different initial cell densities were impacted. Indeed, populations in parafilm-sealed tubes grew faster than in those without sealing (compare Fig 2B with Fig 2A; see Fig B in S1 Figs for S288C). Even cultures at low cell densities eventually grew to saturation, although lag time exhibited stochasticity. When cultured in a 96-well plate sharing headspace, all cell densities could grow to saturation within a short duration (Fig 2C), suggesting that volatiles released from high-density cultures could facilitate the growth of low-density cultures. Interestingly, little stochasticity was observed in the lag time at lower cell densities when all cultures shared headspace, suggesting that small differences amid the gaseous environments within culture tubes gave rise to the stochastic growth dynamics (compare light gray trendlines in Fig 2B and

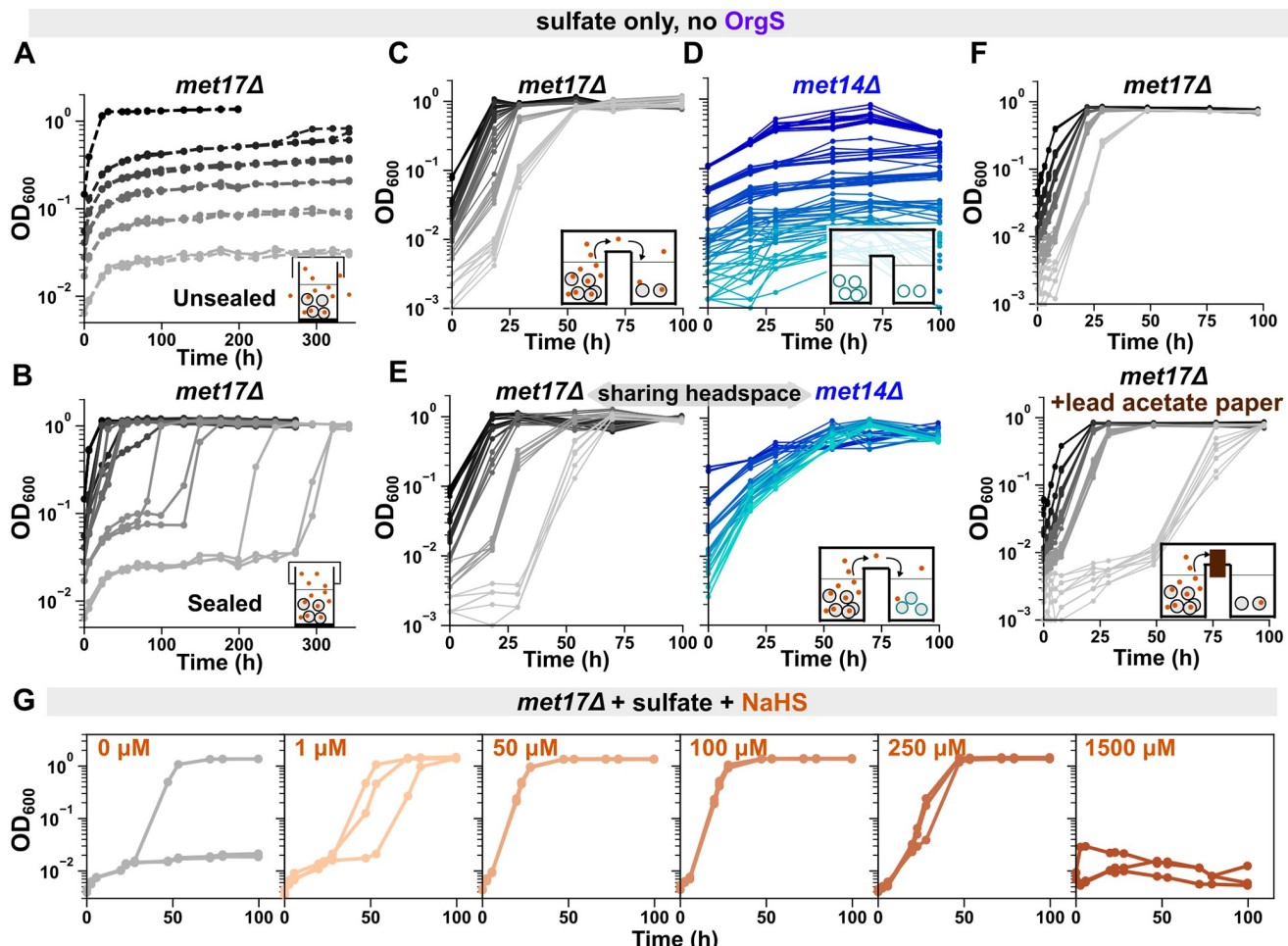

**Fig 2. Hydrogen sulfide gas mediates the density-dependent growth of *met17Δ* on sulfate. (A, B)** Preventing gas escape allows low-density *met17Δ* cultures to grow. Growth curves of *met17Δ* (WY2035) in SD minimal medium, cultured in glass tubes either covered only with plastic lids (**A**, "Unsealed") or additionally sealed with parafilm around the plastic lids (**B**, "Sealed"). Darker shades of gray represent higher initial cell densities. Each trendline represents a 3-ml culture in a tube of 13-mm diameter. At any given initial cell density, 6 identical cultures were initiated, 3 each for the unsealed and sealed treatment. Note that even low-density cultures could eventually grow to saturation when gas exchange was limited by parafilming, indicating that the accumulation of a gaseous growth mediator was enabling population growth. (**C**) Gas from high-density *met17Δ* cultures allowed low-density *met17Δ* cultures in the same plate to grow on sulfate. Growth curves of *met17Δ* (WY2548) at different initial cell densities (indicated by shade of gray) growing in different wells of the same 96-well plate sealed with parafilm. All densities can grow deterministically when the headspace is shared in a multiwell plate. (**D**) At all densities, *met14Δ* only showed residual growth in sulfate. Growth curves of *met14Δ* (WY2539) at different initial cell densities (indicated by shade of blue) growing in different wells of the same 96-well plate. (**E**) Gas from *met17Δ* cultures allowed *met14Δ* cultures to grow on sulfate. Growth curves of *met17Δ* (left panel, gray, WY2548) and *met14Δ* (right panel, blue, WY2539) at different initial cell densities sharing headspace in different wells of the same 96-well plate. (**F**) Lag time of low-density *met17Δ* cultures increased when sulfide was absorbed by lead acetate paper inserted into the gaps between the wells. Growth curves of *met17Δ* (WY2549) at different initial cell densities growing in different wells of the same 96-well plate either with (lower panel) or without (upper panel) lead acetate paper in the gaps. (**G**) Growth of low-density *met17Δ* on sulfate can be promoted by sodium hydrosulfide (NaHS), although high sulfide concentrations are toxic. At each sulfide concentration, each trendline represents one of 3 identical 3-ml liquid cultures (WY2531) in tubes sealed with plastic lids and parafilm. At 1,500 μM, population growth was inhibited. All *met* mutants were in the RM11 background. The data underlying this figure can be found in S2 Data.

2C). In contrast, only residual growth was observed when different initial cell densities of another organosulfur auxotroph, *met14Δ*, were grown on sulfate in a 96-well plate (Fig 2D). The growth dynamics of *met14Δ* were consistent with those observed for the uracil auxotroph, which could not growth at any initial cell density even in 96-well plates (Fig Ci in S1 Figs).

Corroborating previous reports of $H_2S$ release from *met17Δ* cells [15,28], we noted that *met17Δ* cultures growing on sulfate emanated a strong rotten egg odor, and a piece of lead acetate paper placed in the headspace of *met17Δ* cultures turned black (Fig D in S1 Figs). To test whether yeast can uptake and consume ambient $H_2S$ gas, we grew *met14Δ* yeast and *met17Δ* yeast in different wells of the same 96-well plate. Since *MET14* functions upstream of $H_2S$ formation during sulfate assimilation (Fig 1A), we hypothesized that *met14Δ* yeast should grow in the presence of $H_2S$ released by *met17Δ*. Indeed, whereas *met14Δ* alone displayed only residual growth on sulfate (Fig 2D), *met14Δ* cultures at all densities could grow on sulfate when they shared headspace with *met17Δ* growing on sulfate in a 96-well plate (Fig 2E, right panel). Interestingly, low-density *met17Δ* populations showed longer lags in the presence of *met14Δ* (compare Fig 2E, left panel, with Fig 2C), suggesting that the growth dynamics of *met17Δ* cultures were governed by the available quantity of the volatile sulfur forms for which *met14Δ* and *met17Δ* competed. In fact, the growth of low-density *met17Δ* cultures could also be slowed by including lead acetate paper in the airspaces between the wells (Fig 2F), indicating that $H_2S$ from high-density *met17Δ* cultures was driving the growth of low-density cultures.

Congruently, the growth of low-density *met17Δ* cultures was promoted by sodium salts of sulfide over a range of concentrations (Fig 2G and Fig E in S1 Figs). Note that upon acidification, sulfide ions from the salts get protonated to release $H_2S$ gas. At a low sulfide concentration (1 μM), we see stochastic growth dynamics among 3 technical replicates (Fig 2G). In contrast, the 3 replicates grew deterministically at intermediate concentrations of 50 μM and 100 μM (Fig 2G). Thus, the transition from stochastic to deterministic growth dynamics observed with increasing initial cell densities in *met17Δ* cultures (Fig 1D, middle panel) likely resulted from higher sulfide concentrations in high cell-density cultures. Interestingly, cultures could not grow at the highest concentration of sodium hydrosulfide (NaHS) tested (1.5 mM in Fig 2G), indicating that high concentrations of sulfide are toxic to yeast cells, consistent with earlier studies [15,29]. This deterrence to the growth of eukaryotic cells might result from the well-characterized inhibition of mitochondrial cytochrome c oxidase by $H_2S$ [30].

Compared to RM11 *met17Δ*, S288C *met17Δ* required more $H_2S$ to grow and released less $H_2S$. Considerably higher concentrations of sodium sulfide were needed to elicit growth in low-density cultures of S288C *met17Δ* on sulfate (Fig E in S1 Figs, compare ii with i). Moreover, S288C *met17Δ* could grow faster in the vicinity of RM11 *met17Δ* than when growing by themselves in a multiwell plate (Fig F in S1 Figs, compare right panel of iii with ii), suggesting that RM11 *met17Δ* released more $H_2S$ than S288C *met17Δ*. Overall, both lower $H_2S$ release and higher $H_2S$ requirement may contribute to the sluggish growth of S288C *met17Δ* on sulfate compared to RM11 *met17Δ* (compare Fig 1 with Fig A in S1 Figs).

## *HSU1* is required for sulfide assimilation in *met17Δ*

The anomalous growth of *met17Δ* on sulfate indicates the existence of a *MET17*-independent pathway of sulfate assimilation in *S. cerevisiae*. Indeed, *met17Δ* could not grow at any cell density on sulfate-free medium (Fig G in S1 Figs). During sulfate assimilation, Met17 catalyzes the reaction of sulfide with OAH to yield homocysteine (Fig 3A). In vitro, Met17 can also react sulfide with OAS to yield cysteine [19]. Hypothetically, the *MET17*-independent pathway of sulfide fixation could meet the cell's organosulfur requirements by catalyzing either of these reactions (*GENE X* in Fig 3A). To discern if the alternative pathway requires OAH, we examined if *met2Δ* mutants, which lack OAH synthesis [31,32], could also bypass organosulfur auxotrophy. In a 96-well plate, where populations of *met2Δ* at different cell densities shared headspace, we found that the populations only showed residual growth (Fig H in S1 Figs).

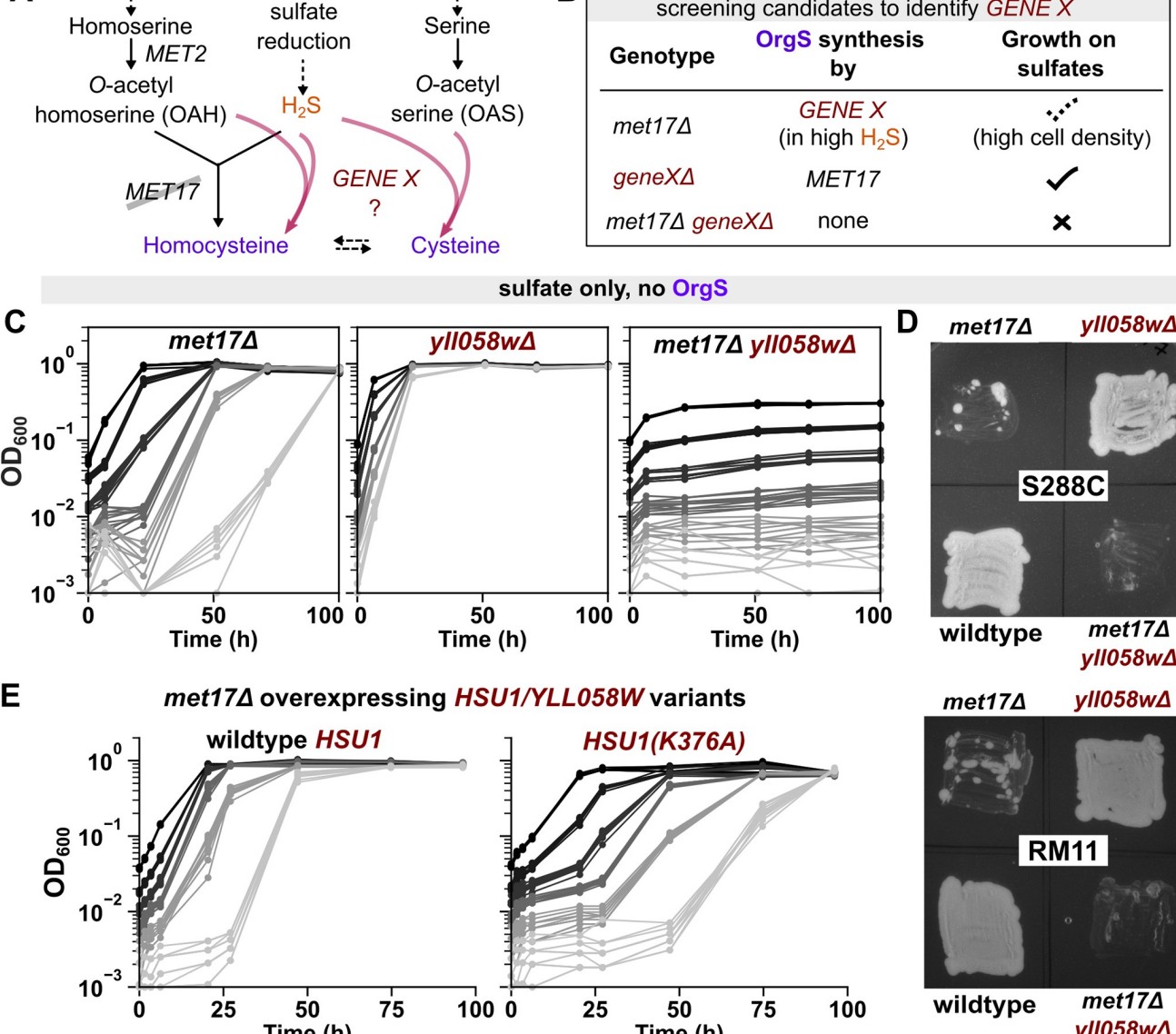

**Fig 3. *HSU1* is required for the growth of *met17Δ* on sulfate.** (**A**) Schematic representing how an unknown *GENE X* could bypass the requirement for *MET17* by performing the enzymatic synthesis of either homocysteine or cysteine in a reaction where hydrogen sulfide is a substrate. Either of the 2 reactions would be sufficient to support the cell's organosulfur requirement, as organosulfurs (purple) are interconvertible. Solid and dashed arrows represent one-step and multistep reactions, respectively. (**B**) Schematic representing the genetic screen to identify *GENE X*. (**C**) *YLL058W* is the hypothesized *GENE X*. Left panel shows that *met17Δ* (WY2590) could grow on sulfate at different initial cell densities when the populations share headspace in a 96-well plate. Middle panel shows that deletion of *yll058w* (WY2597) did not result in an auxotrophy, and all cell densities could grow without lags on sulfate in a 96-well plate. Right panel shows that double deletants of *met17* and *yll058w* (WY2595) could no longer grow on sulfate at any density in a 96-well plate. Strains were of S288C background. (**D**) Unlike *met17Δ* (WY2590, WY2548), the double mutants *met17Δyll058wΔ* (WY2595, WY2642) did not show papillae when patched onto agar plates containing SD medium. *yll058wΔ* (WY2597, WY2639) grew dense lawns similar to wild-type yeast (WY2601, WY2641). For each strain background, the 4 genotypes were patched onto the same agar plate, which was sealed with parafilm and imaged after 5 days. (**E**) The catalytic activity of *HSU1/YLL058W* modulates the growth dynamics of *met17Δ* on sulfate. RM11 *met17Δ* yeast carrying plasmids that overexpressed either the wild-type version of *HSU1* (left panel, WY2645) or the catalytically dead variant *HSU1(K376A)* (right panel, WY2648) were grown on sulfate at different initial cell densities in 96-well plates. Shorter lag times were observed with the overexpression of active *HSU1* compared to inactive *HSU1*. Note that plasmid-carrying strains had a lower exponential growth rate than untransformed *met17Δ*, so the growth dynamics in (**E**) cannot be compared with previous *met17Δ* growth curves. The data underlying this figure can be found in S3 Data.

Thus, it is likely that, similar to *MET17*, the alternative mechanism also utilizes OAH and sulfide.

Met17 is a pyridoxal phosphate–dependent enzyme. We proposed that any protein that performs a catalytic function similar to Met17 will bear structural similarity to the Met17 enzyme. Five candidate genes were identified by a protein BLAST using Met17's sequence: *CYS3*, *STR3*, *ICR7*, *YLL058W*, and *YHR112C*. If any of these genes was *GENE X*, then the *met17Δ geneXΔ* double mutant would no longer grow on sulfate even at a high initial cell density (Fig 3B). However, the *geneXΔ* single mutant should grow on sulfate due to the activity of *MET17* (Fig 3B). Out of the candidates, *cys3Δ* did not meet this criterion of our screen since it is a known cysteine auxotroph [33]. As a quick method of screening the remaining four candidates, we leveraged the fact that *met17Δ* is a background mutation in the yeast deletion library derived from the strain BY4741 of the S288C background [23]. For each candidate, different initial cell densities of the same mutant were inoculated into different wells of the same 96-well plate. This setup allows us to clearly distinguish between residual growth (e.g., *met14Δ*; Fig 2D) and growth to saturation (e.g., *met17Δ*; Fig 2C). Out of the candidates tested, only deletion of *YLL058W* abrogated growth of *met17Δ* on sulfates (Fig I in S1 Figs).

To confirm that the *yll058wΔ* single mutant is not auxotrophic, we deleted the gene in the S288C strain background. Indeed, while *met17Δ* grew in a density-dependent fashion (Fig 3C, left panel), *yll058wΔ* yeast could readily grow on sulfate without supplementation of organosulfurs (Fig 3C, middle panel, compare with prototroph in Fig Cii in S1 Figs). In contrast, double mutants of *met17Δyll058wΔ* could no longer grow to saturation on sulfate at any density (Fig 3C, right panel, compare with uracil auxotroph in Fig Ci in S1 Figs). The double mutants did not show papillae even when patched onto the same solid SD medium plate as $H_2S$-releasing *met17Δ* (Fig 3D) and could not grow even when cultured in the same 96-well plate as *met17Δ* (Fig J in S1 Figs), indicating that the inability of *met17Δyll058wΔ* to grow on sulfate resulted from defects in the assimilation of sulfide.

While we were investigating this phenomenon, 2 other groups simultaneously reported this sulfide-assimilation function of *YLL058W* [26,27], with Yu and colleagues naming the locus *Hydrogen Sulfide Utilizing-1* (*HSU1*) [27]. We will henceforth use the name *HSU1* to refer to *YLL058W*. These 2 groups additionally demonstrated that Hsu1 protein can catalyze the same biochemical reaction as Met17, albeit at considerably lower efficiency [26,27]. In accordance with this purported biochemical activity, we observed that overexpression of functional *HSU1* led to lower lag times in low-density *met17Δ* cultures as compared to those observed with the overexpression of a catalytically dead mutant *HSU1(K376A)* [26] (Fig 3E).

## *HSU1* has a pleiotropic role in sulfur assimilation in wild-type yeast

Based on the fact that *HSU1* can assimilate sulfide when sulfide concentrations are high, we hypothesized 2 possible functions for *HSU1*: (1) an alternate pathway to maximize sulfur assimilation when cells experience sulfur starvation; and/or (2) a mechanism to neutralize excess sulfide when ambient sulfide concentrations get dangerously high (e.g., Fig 2G, last panel). Correspondingly, gene expression might be triggered either by sulfur starvation or by exposure to high sulfide concentrations. To test these hypotheses, we introduced a GFP coding sequence before the stop codon on the C-terminus of *HSU1* in the S288C and the RM11 background. We then imaged these cells after treatment with sulfate-free medium (sulfur-starved in Fig 4A and 4C) or after exposure to different concentrations of sodium sulfide (Fig 4B and 4D). In both backgrounds, exponentially growing wild-type cells showed little GFP signal (Fig 4A and 4C, left panels). Interestingly, a sharp increase in GFP fluorescence was observed under sulfur starvation (Fig 4A and 4C, right panels). Nonspecific starvation may not induce

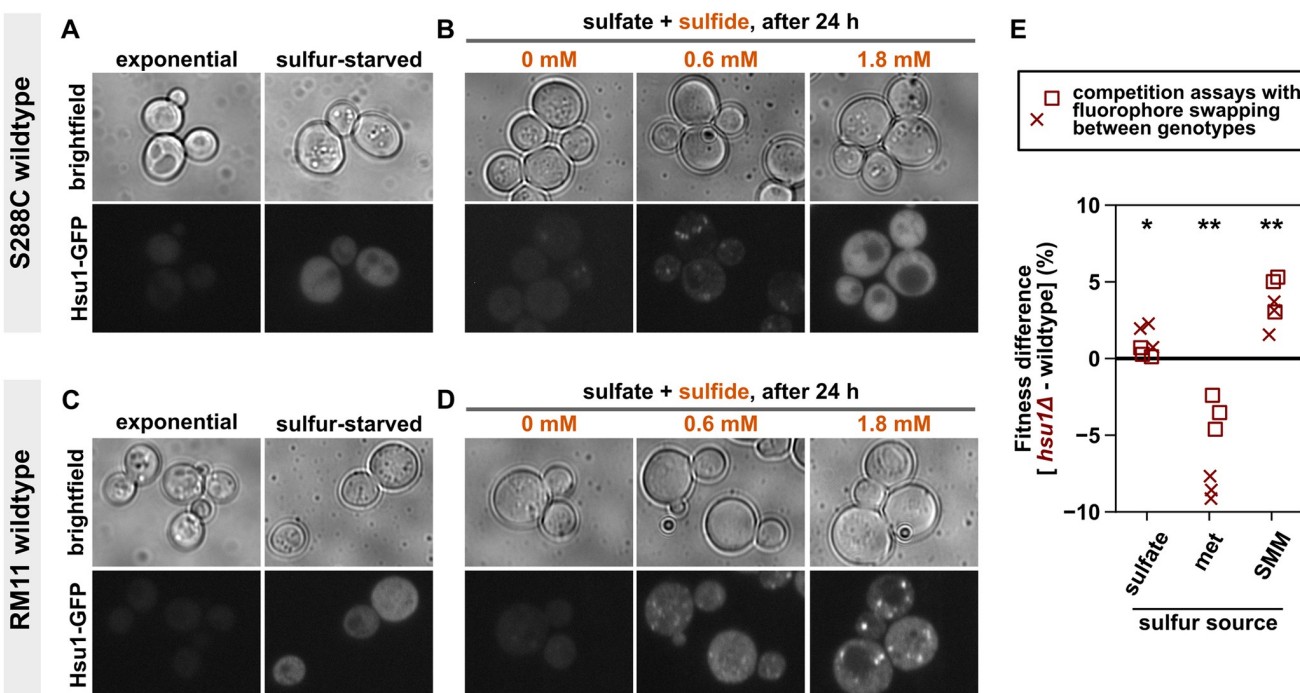

**Fig 4. *HSU1* has a pleiotropic role in sulfur assimilation in wild-type yeast.** The expression profile of Hsu1-GFP in wild-type yeast of S288C (WY2616; **A, B**) and RM11 (WY2618; **C, D**) backgrounds suggest multiple functions of *HSU1* in sulfur metabolism. (**A, C**) Sulfur starvation induced expression of Hsu1-GFP in wild-type yeast within 4 hours. Hsu1-GFP was not detectable in exponential-phase yeast growing in SD minimal medium (left columns). (**B, D**) Adding sulfide (Na₂S) to SD medium also induced expression of Hsu1-GFP in wild-type stationary-phase yeast. Localization of Hsu1 was largely diffused throughout the cytoplasm, although puncta could be observed under some conditions of sulfide exposure. Note that 1.8 mM sulfide impaired growth of wild-type yeast (Fig Mi in S1 Figs), but cells largely remained viable after 1 day of treatment (Fig Mii in S1 Figs). (**E**) *hsu1Δ* have a fitness disadvantage during exponential growth on methionine (met) as the sole sulfur source, and an advantage during growth on S-methylmethionine (SMM). Genotypes were grown to exponential phase in monocultures either in SD medium (containing sulfate) or in sulfate-free medium supplemented with SMM or met and then combined at an approximately 1:1 ratio in the relevant sulfur environment. Fitness difference represents the growth rate difference between *hsu1Δ* and wild type (calculated from flow cytometry–based coculture competition) as a percentage of the overall population growth rate (estimated from OD measurements). To discount any biases arising from the choice of fluorescent proteins, competition assays were performed with fluorophore swapping: crosses represent WY2652 (*hsu1Δ*) against WY1870 (wild type), and squares represent WY2640 (*hsu1Δ*) against WY1810 (wild type). Each assay had 3 technical replicates. All strains were RM11. Ratio dynamics are provided in Fig P in S1 Figs. In all conditions, the fitness effects of *hsu1Δ* are significant according to a one-sample two-tailed *t* test (* denotes $p < 0.05$; ** denotes $p < 0.01$) as well as an alternative test that does not assume that all data points within a condition are identically distributed (Methods: Statistical tests). The data underlying this figure can be found in S4 Data.

Hsu1, as cells grown to stationary phase in SD medium have little Hsu1 protein (Fig 4B and 4D, left panels). When SD medium is supplemented with an intermediate level of sulfide (0.6 mM), Hsu1 protein is induced—more so in the RM11 than in the S288C background (Fig 4B and 4D, middle panels). At a high sulfide concentration (1.8 mM), which can inhibit the growth of wild-type cells (Fig Mi in S1 Figs), Hsu1 protein is induced in both backgrounds (Fig 4B and 4D, right panels). Hsu1 protein generally shows a diffused localization in the cytoplasm, although punctate localization was sometimes observed under sulfide exposure (Fig 4B and 4D, middle and right panels).

To identify the functional significance of *HSU1*, we compared the growth rates of wild-type and *hsu1Δ* S288C yeast under various conditions, including sulfur starvation and sulfide exposure. Monocultures of *hsu1Δ* and wild type displayed comparable exponential growth rates in SD medium (Fig K in S1 Figs). In competition assays under sulfur starvation, no trend could be detected in the growth rate difference between wild type and *hsu1Δ*, and high variation was observed in the outcome (Fig L in S1 Figs). High sulfide concentrations did impair growth, as

was previously reported for *met17Δ* (Fig 2G), but the responses of wild type and *hsu1Δ* to high sulfide were very similar (Fig M in S1 Figs). Competition assays could not be performed for sulfide exposure since sampling would interfere with gas concentrations inside the tube. Since sulfur metabolism is entangled with detoxification of heavy metals in yeast [34], we also compared *hsu1Δ* and wild type under cadmium exposure. Cadmium reduced the growth rate of both genotypes to a comparable extent, and no advantage of *HSU1* could be detected (Fig N in S1 Figs). Thus, *HSU1* did not provide any discernable advantage to wild-type yeast under either sulfur starvation or under stresses such as sulfide or cadmium exposure.

Since *HSU1* was expressed in sulfur-starved cells, we hypothesized that *HSU1* may confer an advantage in the utilization of sulfur sources once starvation is relieved. Recently, quantitative trait loci mapping of wine strains revealed that polymorphisms in *HSU1* affect the production of volatile dimethyl sulfide from the organosulfur S-methylmethionine (SMM) [35]. Yeast usually obtain SMM from plant sources (e.g., from grape juice) and can grow on SMM as the sole sulfur source [36]. We therefore tested the utilization of SMM alongside that of methionine, the more commonly used organosulfur in lab media. For both *hsu1Δ* and wild-type yeast, growth was slower on SMM than on methionine as the sole sulfur source (Fig O in S1 Figs). Notably, in coculture competition experiments, exponentially growing *hsu1Δ* were consistently worse at utilizing methionine than wild type (Fig 4E). Surprisingly, *hsu1Δ* had an advantage when utilizing either SMM or sulfate as the sole sulfur source (Fig 4E). Thus, *HSU1* has a pleiotropic function in the assimilation of different sulfur sources in wild-type yeast.

## A mathematical model assuming low efficiency of sulfide assimilation can explain density-dependent growth of *met17Δ*

What mechanisms might give rise to the positive density dependence observed in the growth of *met17Δ* on sulfate? One possibility is stochastic cell-state switching, which has been widely observed in microbes [37]. Specifically, in the case of *met17Δ* (Fig 5A), all cells may start out being "inactive," i.e., incapable of assimilating sulfide for growth. However, the accumulation of sufficient sulfide may trigger a switch to an "active" or growth-competent state where cells can fix sulfide and give birth to new cells, which, in turn, release more sulfide. A sulfide-dependent regulation of *HSU1*'s expression may serve as the mechanism for such a switch. If this were the case, we would expect Hsu1 protein to be absent or weakly expressed in cells of low-density *met17Δ* cultures that fail to grow. However, considerable Hsu1-GFP expression was observed in low-density *met17Δ* cultures within 4 hours of transfer to SD minimal medium, even if the cultures did not eventually grow (Fig 5B). *HSU1* expression in *met17Δ* probably occurs because organosulfur starvation in *met17Δ* leads to a similar molecular response as sulfur starvation in wild-type cells [38], and we have shown that Hsu1 is induced in the latter condition (Fig 4A and 4C). Thus, if a cell-state switch for sulfide assimilation exists, it is not modulated at the level of Hsu1 protein expression.

A cell-state switch could, however, operate by other molecular mechanisms. For instance, a posttranslational modification might activate Hsu1 once the ambient sulfide concentration has become sufficiently high. Independent of molecular mechanisms, we tested the existence of a cell-state switch by taking cells from a *met17Δ* culture that had started to grow on sulfate and using them to reinitiate cultures at different initial cell densities in SD medium (containing sulfate). We hypothesized that since Hsu1 has already been activated, the fresh inoculations would be able to grow at lower initial densities. However, this was not observed (Fig 5C), indicating that there is no switch to a growth-competent state in *met17Δ* cells or that the switch does not persist through dilution. Thus, a $H_2S$-dependent growth switch is unlikely to be the

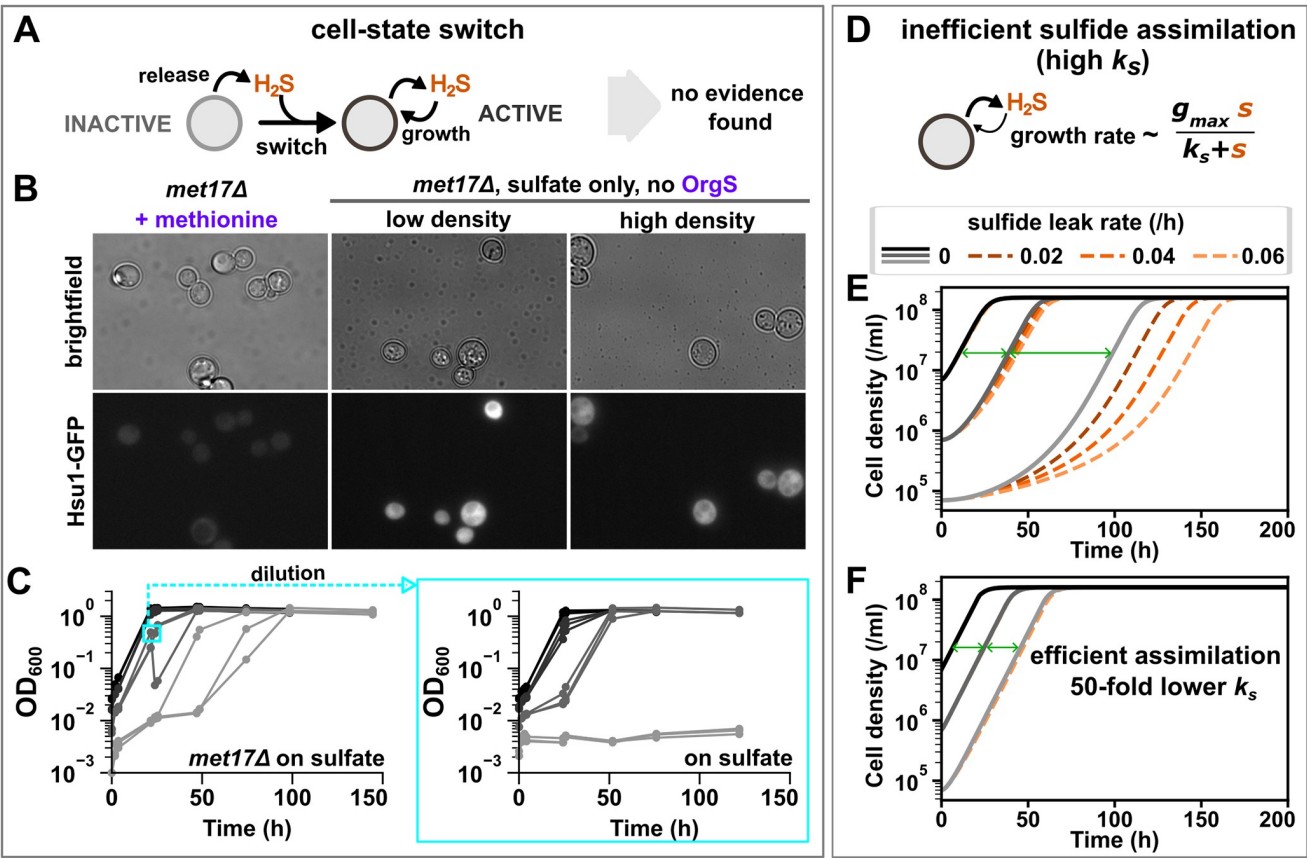

**Fig 5. Inefficient sulfide assimilation can lead to density-dependent growth of *met17Δ*.** (**A**) One possible mechanism for density-dependent growth is that *met17Δ* cells switch to a growth-competent (sulfide-assimilating) state upon sufficient sulfide accumulation. (**B**) Expression of *HSU1* cannot serve as the mechanism of a cell-state switch. While *met17Δ* (WY2623) growing exponentially in methionine-supplemented SD medium did not show much expression of Hsu1 protein, cells in both low-density and high-density cultures showed expression within 4 hours of transfer to SD medium (without methionine or other organosulfur supplements). (**C**) *met17Δ* cells (WY2548) that had started to grow on sulfate continued to show density dependence when diluted in fresh SD medium. This suggests that there is no switch to a growth-competent state in *met17Δ* cells or that the switch is very transient and does not persist dilution. Darker gray shades indicate higher initial cell densities. Each trendline represents a 2.5-ml culture in a parafilm-sealed 13-mm glass tube. Note that since experiments were initiated with exponentially growing cells, longer lags at lower cell densities were the hallmark of density-dependent growth. The data underlying this plot can be found in S5 Data. (**D-F**) Inefficient sulfide assimilation by Hsu1 can give rise to density-dependent growth. (**D**) All *met17Δ* cells are growth competent, with the key assumption being that cell growth rate has a Monod dependence on sulfide concentration. (**E**) A mathematical model (see main text) could generate density-dependent growth dynamics using parameters measured in or inferred from experiments. Lags (green double arrows) were longer at lower cell densities (lighter gray shades). Lag times at lower cell densities were also sensitive to loss or leakage of sulfide (dashed lines). (**F**) Both longer lags and the sensitivity to sulfide loss disappeared when the efficiency of sulfide assimilation is improved. The sulfide concentration corresponding to half-maximal growth rate ($k_s$) was lowered by 50-fold to simulate better sulfide assimilation. Initial cell densities in simulations for (**E, F**) were equivalent to an $OD_{600}$ of 0.001 (light gray), 0.01 (gray), and 0.1 (black), respectively. The code used to generate (**E, F**) can be found in doi.org/10.5281/zenodo.10142030.

sole mechanism producing density-dependent growth of *met17Δ* on sulfate. Similar results were observed for *met17Δ* of S288C background (Fig Q in S1 Figs).

Alternatively, density dependence could simply result from inefficient assimilation of sulfide by Hsu1. To test this hypothesis, we developed a mathematical model describing how *met17Δ* grow by releasing and consuming sulfide. Note that we were able to use a deterministic model instead of a stochastic one because our data suggested that the observed stochastic growth dynamics resulted from experimental factors, rather than biological ones: Low-density *met17Δ* populations only show stochastic lag times in individual tubes (Fig 2B), and not when they share headspace in a multiwell plate (Fig 2C). This suggests that the variance in lag times

results from small differences in the gas environments that the cells experience in each tube, which, in turn, could result from minor defects in sealing.

The model, comprising 2 differential equations describing the dynamics of population density $x$ and sulfide concentration $s$, assumes that (i) all cells are capable of growth, with a carrying capacity $K$; (ii) growth rate has a Monod dependence on sulfide concentration (Fig 5D), with maximal growth rate $g_{max}$ and Monod constant (sulfide concentration at which half-maximal growth rate is achieved) $k_s$; and (iii) sulfide is produced by cells (via sulfate reduction) at a constant rate $r$, consumed by cells at $c$ amount per birth, and lost at a rate of $\delta$. The equations are as follows:

$$\frac{dx}{dt} = g_{max} \frac{s}{k_s + s} \left[1 - \frac{x}{K}\right] x$$

$$\frac{ds}{dt} = rx - c\frac{dx}{dt} - \delta s$$

Some parameters were directly measured in experiments: $c$ = 3 fmole/cell (Fig R in S1 Figs), $K = 1.6 \times 10^8$ cells/ml; $g_{max}$ = 0.26/h (Methods: "Mathematical model"). Others were inferred by fitting the model to the experimental data: $r$ = 0.39 fmole/cell/h and $k_s$ = 7.1 μM (S1 Text, section 3). This $k_s$ includes a factor that accounts for liquid–gas partitioning of sulfide (S1 Text, section 4). Note that parameters measured in in vitro enzyme assays of Hsu1 (e.g., Michaelis–Menten constant) are not applicable to a model of cell growth rate as a function of sulfide concentrations (see Methods "Mathematical model" for further explanation).

This model reproduced the 2 main features of density-dependent growth of *met17Δ* on sulfate: (1) longer lag times at lower cell densities than those at higher densities; and (2) larger variability in lag times at lower cell densities (compare Fig 5E with Fig 2B). Firstly, for 3 cultures separated by equal (10-fold) differences in initial densities, exponential growth curves will have an equal temporal distance in a semi-log plot (for instance, see "prototroph" in Fig 1D). Yet, for *met17Δ*, the spacing between the lower-density curves is longer than that between the higher-density curves (Fig 5E, green double arrows). Secondly, our model predicts that low-density cultures are more sensitive to loss of sulfide (Fig 5E), potentially explaining the stochastic lag times observed in culture tubes (Figs 1D, "*met17Δ*," and 2B). In contrast, small variations in initial cell densities, such as those attributable to pipetting error, had a negligible effect on growth dynamics (S1 Text, section 3), presumably because the lowest cell density in our experiments was already quite large (approximately $10^5$ cells/ml). Finally, the efficiency of sulfide utilization modulates density-dependent growth: Lowering the Monod constant $k_s$ by 50-fold abrogated not only lag at all cell densities but also the sensitivity to sulfide loss (Fig 5F). Thus, low efficiency of sulfide assimilation may be sufficient to explain the density-dependent growth we observe for *met17Δ* on sulfate.

## Discussion

In this study, we have investigated how budding yeast cooperatively overcome the metabolic defect caused by deletion of *MET17*. The enzyme Met17 was believed to be required for yeast to generate organosulfurs from inorganic sulfate [25,39]. However, we have shown that *met17Δ* is not a true organosulfur auxotroph and can, in fact, grow using sulfate as the sole sulfur source, depending on cell density and the extent of gas escape in the experimental setup (Fig 6). This phenomenon occurs because the locus *YLL058W/HSU1* can take over Met17's function of organosulfur synthesis, albeit only at relatively high sulfide concentrations. Specifically, after reducing sulfate to sulfide, *met17Δ* cells fail to react sulfide with OAH to form

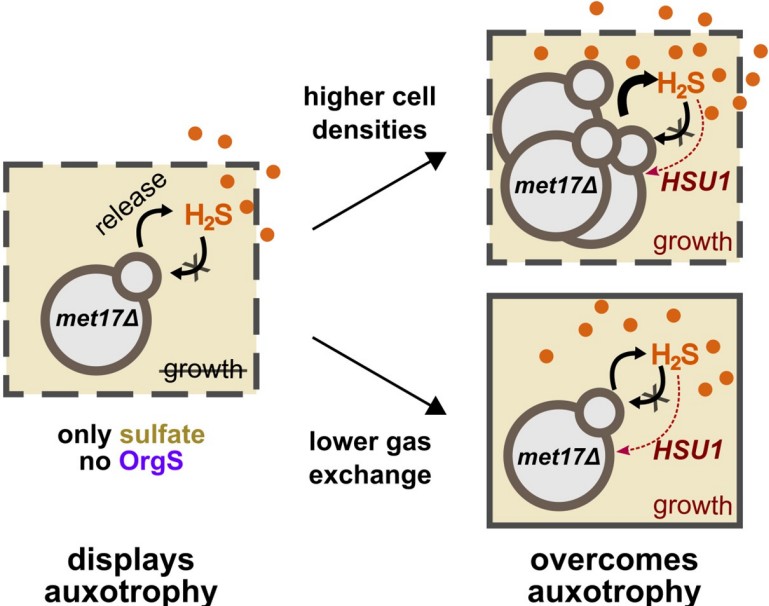

**Fig 6. *met17Δ* can overcome their auxotrophy given sufficient cell densities and/or low gas loss.** Schematic summarizing the results of the study. When low cell-density cultures of *met17Δ* are provided sulfate as the sole sulfur source in sulfide-permeable setups, the yeast are more likely to display auxotrophy. This is because even though *HSU1* is expressed, it does not have access to sufficient $H_2S$ to support cell growth through its inefficient organosulfur synthesis activity. However, either increasing cell densities or lowering gas escape can increase ambient $H_2S$ concentrations to levels where *HSU1*'s weak activity is able to support population growth.

homocysteine. The accumulating sulfide readily crosses cell membranes and partitions between the liquid and air phases. While *HSU1* is expressed in *met17Δ* experiencing organosulfur starvation (Fig 5B and Fig Qi in S1 Figs), it is likely that Hsu1 is inefficient at sulfide assimilation [26,27]. Therefore, ambient sulfide levels need to be considerably high before Hsu1 can synthesize sufficient organosulfurs to fuel cell division. Cultures starting at low cell densities take longer to reach a sufficient level of sulfide or may not reach it at all as sulfide can be lost due to gas exchange or oxidation. Thus, depending on the starting cell density and other experimental factors (e.g., the extent of aeration), *met17Δ* can either grow deterministically, stochastically, or not grow at all on sulfate (Figs 1 and 2). The stochastic growth dynamics of low-density *met17Δ* culture is unlikely to be caused by a sulfide-induced cell-state switch (Fig 5A–5C). Instead, low activity of Hsu1 [26,27] results in density-dependent growth (Fig 5D–5F). Furthermore, at low initial cell densities, growth is particularly sensitive to variations in ambient $H_2S$ gas levels (Fig 5E), which can explain the stochasticity observed in the growth outcome of *met17Δ* on sulfate.

## Reconciling discrepancies in experiments in yeast genetics and metabolism

In retrospect, the erratic growth behaviour of *met17* mutants in the absence of organosulfurs had been noticed prior to our study. Even though *MET17*, *MET15*, and *MET25* were identified as the same locus [13–15], mutants of some *met25* and *met17* alleles failed to grow unless supplemented with organosulfurs [14,25], whereas mutants of other *met17* alleles could grow when supplied with sulfite or sulfide [25]. A careful inspection reveals that even in the study where Cost and Boeke proposed *met15* as a selection marker, *met15* mutants exhibited some low level of growth when patched onto sulfate-containing minimal medium, which lacks

organosulfurs [40]. Finally, when selecting for methyl mercury resistance, *met15* (and *met2*) mutants grew despite the medium not having any organosulfurs [41,42]. While some of these observations could be due to reduced-function mutations associated with specific alleles, it is likely that the leaky auxotrophy of *met17* mutants that we have elucidated in our study could explain some of these perplexing observations.

During the course of this study, 2 other studies were published reporting the anomalous growth of *met17Δ* and identifying the locus *YLL058W/HSU1* as the cause [26,27]. However, the 2 studies disagreed on the observed effect of sulfide on *met17Δ*. Van Oss and colleagues claimed that sulfide accumulation was toxic to *met17Δ* cells, and, therefore, *met17Δ* could grow only once *HSU1* became active and neutralized some of the sulfide [26]. In stark contrast, Yu and colleagues claimed that sulfide accumulation facilitated *HSU1*-dependent growth of *met17Δ* [27]. Consistent with Yu and colleagues' study, we found that a range of sulfide concentrations can promote the growth of both S288C *met17Δ* and RM11 *met17Δ* on sulfate (Fig 2G and Fig E in S1 Figs). Consistent with Van Oss and colleagues' study, we found that high sulfide concentrations can be toxic to yeast cells, but we only observe such growth inhibition with addition of extraneous sulfide (Figs 2G and Figs E and M in S1 Figs). In addition, facilitating gas accumulation by sealing the tubes promoted the growth-propensity of *met17Δ* (Fig 2A and 2B), and removing sulfide using lead acetate paper slowed the growth of low-density *met17Δ* cultures in 96-well plates (Fig 2F), indicating that *met17Δ* usually experienced favorable sulfide regimes in our experiments. We can think of 2 possible explanations for the disparate results in Van Oss and colleagues' study. First, the growth-promotion effect of the sulfide chelator Fe(III)-EDTA may not result from a reduction in sulfide levels. For instance, instead of removing sulfide, the chelator might have improved bioavailability by concentrating the sulfide, or growth might have been promoted simply by Fe(III)-supplementation. Second, despite being closely related, the FY4 strain used in Van Oss and colleagues' study could differ from the S288C strain used in Yu and colleagues' study (and our study) in sulfide release rate or in sensitivity to sulfide. We have noted that RM11 and S288C yeast differ on these parameters (compare Fig 1 with Fig A in S1 Figs; Figs E and F in S1 Figs). It is thus possible that in the experimental setup used in Van Oss and colleagues' study, their yeast strains experience unfavorable sulfide regimes. Either scenario would explain their findings, and we cannot currently distinguish between these possibilities.

Consistent with Yu and colleagues' study [27], we also detect a small advantage for *hsu1Δ* in abundant sulfate (Fig 4E). However, while they reported a small defect for *hsu1Δ* under sulfur limitation [27], we found no consistent fitness defect under sulfur starvation (Fig L in S1 Figs). Although sulfur limitation is different from sulfur starvation, limitation is not well controlled without using chemostats. Additionally, Yu and colleagues used colony counting, a method that produced data with high variance in our hands (Fig L in S1 Figs). This variance arises from the stochasticity associated with plating small numbers of colonies, as well as errors in counting colonies (either manually or using automation). This makes it difficult to detect small fitness differences through colony counting without a high number of replicates. Our analysis, which is based on flow cytometric counting of tens of thousands of cells, has considerably higher precision. Interestingly, we found a pleiotropic role of *hsu1Δ* during exponential growth on methionine, sulfate, or the plant-derived organosulfur SMM. This leads us to speculate that *HSU1* expression during sulfur starvation may confer an advantage subsequently when sulfur sources such as methionine become available. To unravel the complex role of *HSU1* in sulfur assimilation, future competition assays would need to be performed under controlled environments of these different sulfur sources.

Overall, our work has implications for yeast genetics experiments. *met17Δ* is a commonly used selection marker, featuring as a background deletion in the yeast deletion collection [23]

and the GFP-tagged collection [43]. An understanding of the mechanism of its leaky auxotrophy will allow researchers to design protocols that circumvent the caveat. For instance, *met17Δ* can still be used as a selection marker if patching is done at low cell densities and growth phenotypes are assessed within short time intervals.

## Considerations for quantitatively investigating volatile-mediated microbial interactions

Mathematical modeling was useful in providing insights into the mechanisms behind *met17Δ*'s density-dependent growth on sulfate (Fig 5). Our model was shaped by experimental findings and subsequently reproduced the 2 key features of density-dependent growth of *met17Δ*—longer lags and stochastic lag times at lower cell densities (e.g., Fig 2B). We considered 2 biologically relevant mechanisms that could give rise to positive density dependence: (1) a cell-state switch (Fig 5A); and (2) inefficient sulfide assimilation (Fig 5D). We experimentally tested predictions from the cell-state switch hypothesis and failed to find evidence to support it (Fig 5B and 5C). We then modeled a system based on inefficient sulfide assimilation and simulated the experiment of starting at different initial cell densities in sealed glass tubes (Fig 5E). Without considering sulfide loss, the model already reproduced the longer lags observed for low-density *met17Δ* cultures (Fig 5E, gray shades). By including different extents of sulfide loss (Fig 5E, orange shades), we could reproduce high variation in lag times at low cell densities, thus providing an explanation for tube-to-tube variability ("stochasticity") in lag times. We thus achieved a qualitative agreement between the model and experiments. Furthermore, because the parameters used in the models are either explicitly measured or inferred from experiments (Fig 2G), we achieved a fair degree of quantitative agreement: The lag times in Fig 5E are similar in magnitude to those observed in low-density cultures of Fig 2B.

Quantitative agreement can, however, be challenging to achieve due to the volatile nature of the growth substrate $H_2S$, which lends high variability to experimental observations. Our work raises awareness toward 2 factors in particular: First, volatile nutrients can be exchanged between seemingly unconnected microbial populations. By showing that *met14Δ* yeast can grow in the vicinity of sulfide-releasing *met17Δ* (Fig 2E), we demonstrate that assessing auxotrophy in a setup where multiple mutants share headspace can lead to erroneous conclusions. For example, $H_2S$ from neighboring unsealed plates could explain the surprising repeated growth of wild-type FY4 in the absence of inorganic or organic sulfur observed by Van Oss and colleagues (Fig 2D of [26]). Second, small differences in experimental setups could give rise to considerably different experimental outcomes. A mathematical model describing the growth of *met17Δ* on sulfate reveals that, at lower cell densities, growth outcome is sensitive to loss of sulfide (Fig 5E). While loss of nonvolatile substrates primarily occurs through chemical degradation, sulfide could also be lost due to gas escape from the culture chamber, which, in turn, is affected by experimental factors such as temperature [27], the thoroughness of tube sealing, and the use of agitation during growth. The problem is further exacerbated by the nonmonotonic effect of sulfide on *met17Δ*: growth-promotion at lower doses and growth-inhibition at higher doses. Growth outcomes are thus sensitive to the exact sulfide environment that the yeast experience, which is difficult to quantitatively compare between different studies.

The liquid–gas partitioning of sulfide could pose an additional challenge for quantitatively comparing results from different setups. However, this hindrance may not be as severe as one would naively imagine. If the timescale of liquid–gas partitioning is much faster than the biological processes (e.g., consumption and release) and gas leakage, then partitioning does not need to be modeled explicitly (S1 Text, section 4). Rather, partitioning can be accounted for by including a liquid–gas partition factor in the Monod constant ($k_s$) and the leakage rate ($_\delta$).

## Possible functions of *HSU1* in sulfur metabolism

Multiple lines of evidence indicate *HSU1*'s involvement in sulfur metabolism. First, *HSU1* transcription is induced during sulfur limitation and starvation [44–46]. Hsu1 protein is also induced during sulfur starvation (Fig 4A and 4C) or limitation [27]. Second, in vitro biochemical assays show that, similar to Met17, Hsu1 can act as a homocysteine and cysteine synthase, although Hsu1 is much less active than Met17 [26,27]. Finally, competition assays revealed that *HSU1* confers a growth-rate advantage in utilizing methionine and a disadvantage in utilizing either SMM or sulfate as sole sulfur sources (Fig 4E). This suggests a pleiotropic role of *HSU1* in sulfur assimilation.

Quantitative trait loci mapping has previously implicated *HSU1* in utilizing SMM for the production of the volatile dimethyl sulfide (DMS) [35]. *HSU1* resides on the same chromosomal segment as the SMM permease *MMP1* and the SMM-metabolizing enzyme *MHT1* [47]. It is thus possible that *HSU1* directly catalyzes reaction(s) that convert SMM to DMS. The enzymes methionine gamma-lyases, named so for degrading methionine to the volatile methanethiol, can also catalyze the gamma-elimination of SMM to produce DMS [48]. These enzymes have so far only been found in bacteria and plants [49], but it is worth noting that wine yeast do show increased methanethiol production in methionine-supplemented media [50]. Using a protein BLAST to search for *S. cerevisiae* proteins, which share homology with the methionine gamma-lyase from *Pseudomonas putida*, picks up multiple sulfur metabolism genes including *MET17* and *HSU1* (Fig S in S1 Figs). Interestingly, methionine gamma-lyases are multicatalytic, accepting methionine, SMM, cysteine, OAH, OAS, and other related compounds as substrates for elimination or substitution reactions [49]. The various reported functions of *HSU1* also suggest catalytic flexibility—OAH/OAS sulfhydrylase [26], breakdown of SMM to DMS [35], and roles in methionine and SMM assimilation (Fig 4E). We can therefore speculate that *HSU1* is a methionine gamma-lyase. To establish this classification, biochemical characterization of *HSU1* with various substrates could be carried out in the future. Any in vitro evidence of multicatalysis would then need to be verified in vivo by using isotope tracing coupled with mass spectrometric detection of volatiles in wild type and *hsu1Δ*.

If *HSU1* encodes a multicatalytic enzyme, the fitness effects of the gene could differ when substrates and products vary between different sulfur environments. In accordance, we observed pleiotropic effects when wild type and *hsu1Δ* competed on different sulfur sources (Fig 4E). *HSU1*'s ability to compensate for the lack of *MET17* also varied considerably between the vineyard isolate RM11 and the lab strain S288C (compare Fig 1 with Fig A in S1 Figs). In fact, the *HSU1* gene shows considerable variation across the yeast strains represented in the *S. cerevisiae* Genome Database [51]: Of the 41 strains, *HSU1* contains large deletions in 5 strains, in addition to multiple polymorphic regions (Fig T in S1 Figs). In contrast, *MET17* is highly conserved in all except one of these strains. This variation in *HSU1* does not result purely from its proximity to the telomere since genes such as *GTT2* are more proximal to the telomere but are still highly conserved. Altogether, these data indicate that *HSU1*'s function has diversified, which aligns well with the notions of *HSU1*'s catalytic flexibility and its role in secondary metabolism.

A better understanding of *HSU1* and a probe into yeast methionine gamma-lyases holds immense potential for application. For instance, many volatile sulfur compounds are crucial to wine aroma, but their sources are still poorly understood [52]. Identifying methionine gamma-lyase genes in yeast would enable the wine industry to fine-tune the production of volatiles by engineering the substrate specificity of these enzymes. Novel methionine gamma-lyase genes would also be of interest to human health since these enzymes have long been studied as drug targets in pathogens and for anticancer therapies [49].

In sum, our work provides important considerations for experimental design in both yeast genetics and volatile-mediated microbial interactions, and reveals less understood aspects of sulfur metabolism in yeast with potential applications in therapeutics and winemaking.

## Methods

### Yeast strains

All yeast strains used in this study are listed in S1 Table. Yeast nomenclature follows the standard convention. Primers used in the study are listed in S2 Table.

Deletion strains were constructed either via yeast crosses or by homologous recombination [53,54]. Crosses were carried out by mating parent strains, pulling diploids and sporulating them, dissecting tetrads, and genotyping haploids on suitable selection media. As an example, *hsu1Δ* (WY2597) was constructed by PCR-amplifying the *KANMX* resistance gene from a plasmid (WSB26; [55]) using the primers WSO705 and WSO706, with a 45-base pair homology to the upstream and downstream region of the *HSU1* gene, respectively. A diploid heterozygous for *met17Δ* was then transformed with the PCR product, and transformants were selected on a G418 plate. Successful deletion was confirmed via a checking PCR with a primer upstream of the *HSU1* gene (WSO707) paired with an internal primer for the amplified *KANMX* cassette (WSO144). This diploid was then sporulated, tetrads dissected and replica-plated onto G418 and SD medium plates for genotyping, such that both *hsu1Δ* and *met17Δhsu1Δ* strains could be generated from this protocol. Mating-type genotyping of selected clones was carried out using a PCR-based protocol with primers WSO690-692 [56]. One exception to this standard deletion process was the generation of a marker-free deletion of *MET17* (WY2548) using the counter-selectable marker amdSYM [57].

For generating strains with different fluorescent labels for competition assays, either *HSU1* was deleted in a diploid strain heterozygous for constitutively expressed fluorophores or *hsu1Δ* was crossed to wild-type strains expressing the fluorophore of interest. The fluorophores tagged the constitutively expressed, cytoplasmic protein Fba1. Colonies expressing the fluorophores could be distinguished by visual inspection of colony color for strong fluorophores like mCherry or on a low-magnification fluorescence microscope for weaker fluorophores like BFP. For instance, the RM11 *hsu1Δ* strain WY2640 was generated by deleting *HSU1* in the diploid WY2541, followed by sporulation, tetrad dissection, and genotyping to select mCherry-expressing *hsu1Δ* haploids. WY2640 was then crossed with a BFP-expressing MATα wild type (WY1812), and tetrads were dissected to generate the BFP-labeled *hsu1Δ* haploid WY2652. For S288C, a MATα *met17Δhsu1Δ* strain (WY2602) was crossed with wild-type strains expressing either eGFP (WY1364) or mOrange (WY1376) to obtain *hsu1Δ* haploids expressing either eGFP (WY2608) or mOrange (WY2612).

For some S288C strains, the loxP-flanked *KANMX* cassette, which replaced *HSU1*, was looped out using the *CRE* gene expressed from a plasmid carrying the dominant selection marker ClonNAT (WSB194). WY2608 was transformed with WSB194, and a selected transformant was allowed to grow to saturation in rich medium (YPD). The culture was appropriately diluted to yield approximately 300 colonies on a YPD plate. Colonies were then simultaneously patched onto YPD plates with and without G418. A colony that could grow on YPD but not on the G418 plate (i.e., had successfully looped out the *KANMX* cassette) was further propagated by streaking on YPD. A few colonies were selected and simultaneously patched on YPD plates with and without ClonNAT to select for loss of the *CRE* plasmid. A colony that grew on YPD but not ClonNAT was selected and stored to serve as an eGFP-labeled *hsu1Δ* strain without *KANMX* (WY2637).

For constructing the strain with *HSU1* tagged at the C-terminus with GFP, the *eGFP-KANMX* cassette was amplified from WSB65 [58] using primers WSO710 and WSO706. Wild-type S288C (WY2601) and RM11 (WY1203) were transformed with the cassette, and transformants were selected on a G418 plate. Transformants were confirmed using a checking PCR with primers WSO711 and WSO159 resulting in strains WY2616 (S288C) and WY2618 (RM11). These strains were mated with MATα *met17Δ* strains (WY577 for S288C and WY2533 for RM11). Dissected tetrads were screened on an SD medium plate as well as a G418 plate to select *met17Δ* spores with *HSU1-eGFP* for each strain background (WY2620 and WY2623).

*HSU1* overexpression strains (Fig 3E) were constructed by transforming WY2549 with plasmids pARC0172 and pARC0245 received from the Carvunis lab [26].

## Media, chemicals, and culturing conditions

Prior to an experiment, strains were revived from glycerol stocks stored at −80°C by streaking on YPD plates (2% agar included in liquid YPD: 10 g/L yeast extract, 20 g/L peptone, 20 g/L glucose) and incubating at 30°C for 48 hours. Around 2 ml of liquid YPD was then inoculated from a single colony from the plate and grown overnight at 30°C with agitation. These saturated YPD cultures were used to inoculate synthetic minimal medium (SD: 6.7 g/L Difco Yeast Nitrogen Base without amino acids but with ammonium sulfate from Thermo Fisher Scientific, Waltham, MA, United States of America, and 20 g/L D-glucose) supplemented with any amino acids required by the auxotroph used in that experiment. For experiments with *met17Δ*, 20 mg/l of methionine was added. For WY2035, 20 mg/l of methionine and 30 mg/l lysine were added. For strains from the yeast deletion library used in Figs F and I in S1 Figs, 20 mg/l methionine, 20 mg/l uracil, 20 mg/l histidine, and 60 mg/l leucine were added for exponential growth [59]. Strains carrying plasmids for overexpression of *HSU1* variants were revived on G418 plates (YPD plates with 200 mg/l G418). This concentration of G418 was maintained in SD medium throughout the experiment when these strains were used. Sulfur starvation was induced by washing and culturing exponentially growing cells in sulfate-free medium, which was prepared by replacing all sulfate salts in SD medium; a detailed list of ingredients and preparation notes are provided in S1 Protocol.

For sulfide treatment, SD medium was supplemented with sodium sulfide, either NaHS or $Na_2S$. Experiments in Fig 2G and Fig R in S1 Figs used a 1 M NaHS stock solution (gifted by Prof. Mark Roth, FHCRC, Seattle, WA, USA). All other experiments used a 1 M stock solution of $Na_2S$ (Fisher Scientific catalogue no. 10656811), prepared in 0.001 M NaOH in an anaerobic chamber. Both the stock solutions were stored in an anoxic environment in glass tubes with self-sealing septa. Immediately prior to an experiment, approximately 50 to 100 μl of the sulfide stock was drawn out using a 27G-$1_{1/4}$ gauge needle. The necessary amount was pipetted into the cell culture tubes on the glass wall, and tubes were immediately sealed. The sulfide was mixed by vortexing only after sealing to minimize loss of gaseous $H_2S$ during handling.

For Fig 4E and Fig O in S1 Figs, SMM (product no. M0644, Tokyo Chemical Industry, Tokyo, Japan) was added at a final concentration of 0.2 mM to sulfate-free medium. For cadmium exposure (Fig N in S1 Figs), media were supplemented with the stated concentrations of cadmium sulfate 8/3-hydrate (product no. C3266, Sigma-Aldrich, currently Merck KGaA Darmstadt, Germany).

Glass tubes of either 13-mm or 18-mm diameter, with loosely fitted plastic lids, were used for culturing. Culture volume was maintained at either 2.5 ml or 3 ml in the smaller tubes or 7 ml in the larger tubes. Where sealing is mentioned, the tubes were additionally sealed with a layer of cling film and at least 2 rounds of parafilm. Tubes were placed either in a 30°C incubator on a tilted rack with 250 rpm agitation or on a rotor wheel in a 30°C warm room.

## Measuring growth dynamics and growth rate

All experiments were initiated with exponentially growing cells for better reproducibility. On the evening before an experiment, saturated YPD overnights were used to inoculate SD medium supplemented with any necessary amino acids for each strain. The next day, optical density at 600 nm (OD) was tracked to check that cells were growing exponentially. Only cultures between 0.2 and 0.4 readings were used to initiate experiments.

For density dependence assays, cells growing exponentially on SD medium with any necessary supplements were washed thrice in SD medium, before being appropriately diluted to achieve the desired cell densities. The nutrient required to compensate for the auxotrophy of interest was withheld during and after the wash, though other supplements would be maintained throughout. For instance, when the effect of *met17Δ* was assessed in WY2035, which additionally carry a lysine auxotrophy, the medium was supplemented with lysine before, during, and after the wash. For prototrophs, this procedure involved no change of medium. For each starting cell density, a single cell suspension was prepared, and equal volume of this suspension was distributed into sterilized, factory-clean glass tubes.

OD measurements were carried out on a Genesys 20 spectrophotometer (Thermo Fisher Scientific, Waltham, MA, USA) with an adjustable adapter for tubes of different diameters. Each tube was measured thrice with small rotations and the middle value was noted, to avoid the influence of scratches on the glass surface. Any values under 0.001 were below the sensitivity of the machine and were thus replaced by 0.001 for plotting. Note that on this device with tubes of 13-mm diameter, we have estimated that 1 OD corresponds to roughly $7 \times 10^7$ cells. On this device, OD no longer scales linearly with cell density beyond 0.5. Growth curves have not been corrected for this nonlinearity.

For plate experiments, cells were cultured at a volume of 150 μl per well in a Costar 3370 96-well plate (Corning, Glendale, AZ, USA). The plates were maintained in a 30°C incubator, and OD was periodically measured with the lid on using a Biotek Synergy MX plate reader (Agilent Technologies, Santa Clara, CA, USA). Prior to any measurement, the cultures were agitated to suspend cells using a Teleshake magnetic shaking device (Thermo Fisher Scientific, Waltham, MA, USA), and a custom-built "lid warmer" was used to remove condensation from the plate lid [60]. In each plate, the bottom row was filled with sterile SD medium for blanking. Any values under 0.001 were below the sensitivity of the machine and were thus replaced by 0.001 for plotting.

For Figs K, Mii-iii, Nii, and O in S1 Figs, growth rates were measured as the slope of natural log of OD measurements over time. Due to the nonlinearity above OD 0.5 on our spectrophotometer, growth rates were only measured from OD values between 0.1 and 0.5. In Fig L in S1 Figs, flow cytometry or colony counting was used to measure population densities in cocultures. In this case, the difference in growth rates of the 2 populations in a coculture was calculated as the slope of natural log of population ratios over time. To calculate percentage fitness difference (Fig 4E), the growth rate difference of the populations measured by flow cytometry was scaled to the combined growth rate of the coculture estimated from OD measurements.

## Fluorescence microscopy and imaging

Exponentially growing wild-type cells were either washed with sulfate-free medium for sulfur starvation or subjected to different concentrations of sodium sulfide in SD medium. *met17Δ* were grown to exponential phase in methionine-supplemented SD and washed with SD without organosulfur supplements. In all treatments, cultures were set up at an OD of approximately 0.1, volume 2.5 ml, in 13-mm glass tubes. Tubes were sealed with parafilm for sulfide treatment. For imaging at different time points, 4 μl of the cell culture was spread under a

coverslip on a slide. The fluorescence microscopy setup comprised a Nikon Eclipse TE-2000U inverted fluorescence microscope (Nikon, Tokyo, Japan) connected to a cooled CCD camera for fluorescence and transmitted light imaging. Image acquisition was done with an in-house LabVIEW program. Images were captured using either a 100× objective (Fig 4A–4D) or a 40× objective (Fig 5B and Fig Qi in S1 Figs). For imaging GFP, the filter cube used was Chroma 49002-ET-EGFP (exciter: 470/40×, emitter: 525/50m, Dichroic: T495LP; Chroma Technology, Bellows Falls, VT, USA). Imaging conditions and exposure times were kept constant for imaging different treatments. Images of yeast growth on agar plates were captured using a G:BOX F3 gel imager (Syngene, Cambridge, UK). Image processing was done on Fiji [61].

## Flow cytometry

A detailed description has been previously published [62]. Population compositions were measured by flow cytometry using a DxP10 (Cytek, Fremont, CA, USA). To measure absolute cell densities in a sample, fluorescent beads (3-μm red fluorescent beads, Catalogue no. R0300, Thermo Fisher Scientific, Waltham, MA) of known concentration were added to each sample. The bead concentration was independently determined by counting on a haemocytometer. Additionally, in assays where cells were stressed by sulfur starvation, a final 20,000-fold dilution of the ToPro3 (Molecular Probes T-3605; Eugene, OR, USA) was added to each sample to distinguish between dead and live cells [62]. ToPro3-stained dead cells form a distinct cluster from live cells by producing a high signal on the RedFL1 channel (EX: 640 nm, filter: 661/16 BP). Gating and subsequent analysis were done using FlowJo v10.8 Software (BD Life Sciences, Ashland, OR, USA). In brief, beads were first separated from cells based on their high emission on VioIFL1 (EX: 405 nm, filter: 445/50 BP) and BluFL2 (EX: 488 nm, filter: 585/45 BP) channels. The cell cluster was further refined by their scattering profile, and, where ToPro3 was used, dead cells (with high RedFL1 signal) were excluded. Within live cells, GFP- or mCitrine-positive clusters were defined by their BluFL1 (EX: 488 nm, filter: 530/30 BP) emission, mCherry- or mOrange-positive clusters were defined by their YelFL1 (EX: 561 nm, filter: 590/20 BP) emission, and BFP-positive clusters were defined by their VioIFL1 emission.

## Competition and functional assays

For competition assays, *hsu1Δ* and wild-type strains constitutively expressing fluorophore-tagged Fba1 were generated. For competing strains during exponential growth on different sulfur sources (Fig 4E and Fig P in S1 Figs), each strain was grown to exponential phase in either SD medium (sulfate as sulfur source) or in sulfate-free medium supplemented with either methionine or SMM. The growth rates of the monocultures were tracked for 1 to 2 hours before diluting each genotype to OD 0.07 and mixing the 2 genotypes from the same sulfur environment in a 1:1 ratio making up a 3-ml coculture. For cadmium exposure (Fig N in S1 Figs), 40 μM of cadmium sulfate was included in the cocultures in SD medium. For competition under sulfur starvation (Fig L in S1 Figs), strains were grown to exponential phase in SD medium before being washed 3 times with sulfate-free medium. Monocultures were diluted to 0.1 OD and combined in a 1:1 ratio to generate 3-ml cocultures. All competition assays were set up in sterilized, factory-clean 13-mm glass tubes and 3 technical replicates were set up for each test condition in every assay. The cocultures were assessed at different time points, usually 3 to 4 times on day 1 and once a day for subsequent days where applicable. For sampling, ODs were measured, and cells were appropriately diluted to achieve an event rate below 1,000/s on the flow cytometer. Samples were prepared as described in section "Flow cytometer." For experiments in Fig L in S1 Figs, samples were collected for both flow cytometry and colony counting at each time point. To achieve consistency in manual colony counting, samples were

diluted to get roughly 300 colonies on each plate. Cell densities were, however, estimated by OD for plating and were, therefore, not very accurate. In practice, we obtained 150 to 200 colonies on each plate. Cells were spread on 2 YPD plates for each time point and allowed to grow for 2 to 3 days in a 30˚C incubator before counting. A Nikon AZ100 upright microscope was used to distinguish colonies expressing GFP from those expressing mOrange.

As a control in flow cytometry–based competition, any biases arising from fluorophore choice were accounted for by repeating the competition assays with strains where the fluorophores had been swapped between the 2 genotypes. For sulfur starvation and cadmium exposure, the choice of fluorophores made a measurable impact on the outcome of the assay (Figs Li and Ni in S1 Figs). For this reason, the effect of cadmium exposure was assessed by OD measurements in monocultures for each genotype (Fig Nii in S1 Figs).

The impact of sulfide exposure was also assessed by OD-based growth rate measurement in monocultures since sampling for flow cytometry would result in gas leakage. We additionally assessed cell densities with flow cytometry before and 24 hours after sulfide exposure (Fig Mii in S1 Figs, right panel).

## Mathematical model

The basic model and parameter values are described in the main text. The detailed development of the model is provided in S1 Text, along with the inference of some parameters ($r$ and $k_s$) from data-fitting. All associated code is provided in doi.org/10.5281/zenodo.10142030. Sulfide consumption rate $c$ was experimentally determined by measuring the increase in cell density at different sulfide concentrations (Fig R in S1 Figs). Maximum growth rate $g_{max}$ of $met17\Delta$ was measured from the 50-μM and 100-μM datasets in the sodium hydroxide experiment (Fig 2G) since growth was the fastest and deterministic under these sulfide treatments. Growth rate was calculated as the slope of natural log of OD measurements over time. Carrying capacity $K$ was also determined from the NaHS experiment by correcting stationary-phase OD measurements for the nonlinearity of our instrument at high OD values and converting the values to cell densities using the relationship 1 OD = $7 \times 10^7$ cells. Note that data used for parameter inference and measurement correspond to RM11 yeast.

Parameters measured in in vitro enzyme assays of Hsu1 cannot be applied to our model. When the population density is much smaller than the carrying capacity, the growth rate will have a Monod-type dependence on sulfide concentration. While this relationship is similar in form to the Michaelis–Menten enzyme kinetics equation, the 2 equations are not equivalent. The Michaelis constant and the Monod constant, respectively, describe the substrate concentration at which half-maximal enzyme activity and half-maximal cell growth is attained. Cell growth is a function not only of enzyme affinity for substrate but also of additional factors such as the number of enzymes in the cell and the coordination between the reaction and the rest of cellular metabolism. Indeed, Van Oss and colleagues report that the Michaelis constants of both Hsu1 and Met17 for sulfide are in millimolar range [26] but that concentration would be toxic to yeast (Fig 2G and Fig Mi in S1 Figs). Thus, inferring $k_s$ from our own data (as described in S1 Text) is a more reasonable approximation.

## Statistical tests

Where significance testing has been mentioned, the relevant calculations are included in the associated data files. In Fig 4E, a one-sample two-tailed $t$ test was performed by pooling the data points from 2 independent competition assays and testing for a significant deviation from 0. However, this approach may violate the requirement that data be drawn from the same distribution for a one-sample $t$ test. We therefore verified the results with a more involved test

that bypasses this requirement and instead assumes that the genotype and fluorophore have additive effects.

We modeled the expected growth rate of a strain as a linear combination of 2 factors: the genotype (wild type or $hsu1\Delta$) and the fluorophore (mCherry or BFP). For example, the average growth rate of the wild-type strain with mCherry is modeled as:

$$r_{wt,red} = r_{wt} + r_{red}$$

Our average measured growth rate differences may then be written:

$$\delta r_{wt=blue} = r_{hsu1\Delta,red} - r_{wt,blue} = r_{hsu1\Delta} - r_{wt} + r_{red} - r_{blue}$$

$$\delta r_{wt=red} = r_{hsu1\Delta,blue} - r_{wt,red} = r_{hsu1\Delta} - r_{wt} + r_{blue} - r_{red}$$

where $r_{wt}$ and $r_{hsu1\Delta}$ are the average basal growth rates of the wild type and $hsu1\Delta$ backgrounds, and where $r_{red}$ and $r_{blue}$ are the average growth rate effects of mCherry and BFP. By combining the above equations, we have:

$$\delta r_{wt=blue} + \delta r_{wt=red} = 2(r_{hsu1\Delta} - r_{wt})$$

It follows that if $\delta r_{wt=blue} \neq -\delta r_{wt=red}$, then $r_{hsu1\Delta} \neq r_{wt}$. We thus tested the null hypothesis $\delta r_{wt=blue} = -\delta r_{wt=red}$ using a two-tailed two-sample $t$ test with equal variances. Note that absolute growth rate differences (not fitness differences) were used for this test. The resulting $p$-values (0.02, 0.0001, and 0.002, respectively, for the sulfate, methionine, and SMM conditions) verify the results of the simpler one-sample $t$ test.

### Sequence alignments

For the growth screen described in Fig 3B, candidate genes were identified by querying NCBI's protein BLAST with the S288C Met17 protein sequence from the *Saccharomyces* Genome Database [51] and restricting the search to the *S. cerevisiae* S288C nonredundant protein database. Default blastp parameters were used. For Fig S in S1 Figs, the MdeA protein sequence from *Pseudomonas putida* (UniProt ID P13254) was blasted against the *S. cerevisiae* genome, and graphics were generated using UniProt [63]. For Fig T in S1 Figs, protein sequences of Hsu1, Met17, and Gtt2 from 41 *S. cerevisiae* strains were aligned using the strain alignment function in the *Saccharomyces* Genome Database [51], and graphics were generated using Jalview [64].

## Supporting information

**S1 Figs. File containing supplementary figures A-T.**
(PDF)

**S1 Text. File detailing model development and parameter inference.**
(PDF)

**S1 Protocol. Composition and preparation notes for sulfate-free medium.**
(XLSX)

**S1 Table. List of all yeast strains used in this study.**
(XLSX)

**S2 Table. List of all primers used in this study.**
(XLSX)

**S1 Data. Data underlying Fig 1C and 1D.**
(XLSX)

**S2 Data. Data underlying Fig 2A–2G.**
(XLSX)

**S3 Data. Data underlying Fig 3C and 3E.**
(XLSX)

**S4 Data. Data underlying Fig 4E.**
(XLSX)

**S5 Data. Data underlying Fig 5C.**
(XLSX)

**S6 Data. Data underlying Fig A in S1 Figs.**
(XLSX)

**S7 Data. Data underlying Fig B in S1 Figs.**
(XLSX)

**S8 Data. Data underlying Fig C in S1 Figs.**
(XLSX)

**S9 Data. Data underlying Fig E in S1 Figs.**
(XLSX)

**S10 Data. Data underlying Fig F in S1 Figs.**
(XLSX)

**S11 Data. Data underlying Fig G in S1 Figs.**
(XLSX)

**S12 Data. Data underlying Fig H in S1 Figs.**
(XLSX)

**S13 Data. Data underlying Fig I in S1 Figs.**
(XLSX)

**S14 Data. Data underlying Fig J in S1 Figs.**
(XLSX)

**S15 Data. Data underlying Fig K in S1 Figs.**
(XLSX)

**S16 Data. Data underlying Fig L in S1 Figs.**
(XLSX)

**S17 Data. Data underlying Fig M in S1 Figs.**
(XLSX)

**S18 Data. Data underlying Fig N in S1 Figs.**
(XLSX)

**S19 Data. Data underlying Fig O in S1 Figs.**
(XLSX)

**S20 Data. Data underlying Fig P in S1 Figs.**
(XLSX)

**S21 Data. Data underlying Fig Q in S1 Figs.**
(XLSX)

**S22 Data. Data underlying Fig R in S1 Figs.**
(XLSX)

## Acknowledgments

We would like to thank Mark Roth for immensely useful discussions about the chemistry and handling of sulfide, as well as providing us with a stock solution of sulfide. We greatly appreciate the assistance of Olly Jarvis for competition assays, and that of Yuejia Zhu and Michael Gong for generating yeast strains. We are indebted to the Ralser lab for sharing yeast strains and to the Carvunis lab for sharing plasmids for revision experiments; if our work had to be scooped, we couldn't hope for friendlier competition than these two labs. Li Xie, David Skelding, and all members of the Shou lab have been vital in providing critical input to this work. We are additionally grateful to Maitreya Dunham and Dana Miller for discussions on sulfide metabolism, Hanadi Rammu for help with sulfide handling, Stuart Harrison for useful tips on sulfur metabolism, Shawna Miles for yeast protocols, and Linda Breeden and members of the Sue Biggins lab for hosting and supporting us during our transition from Seattle to London.

## Author Contributions

**Conceptualization:** Sonal, Wenying Shou.

**Data curation:** Sonal.

**Formal analysis:** Sonal, Alex E. Yuan, Wenying Shou.

**Funding acquisition:** Sonal, Wenying Shou.

**Investigation:** Sonal, Alex E. Yuan, Xueqin Yang.

**Methodology:** Sonal, Alex E. Yuan, Wenying Shou.

**Project administration:** Sonal, Wenying Shou.

**Resources:** Sonal.

**Software:** Sonal, Alex E. Yuan.

**Supervision:** Sonal, Wenying Shou.

**Validation:** Sonal, Xueqin Yang.

**Visualization:** Sonal, Alex E. Yuan, Wenying Shou.

**Writing – original draft:** Sonal.

**Writing – review & editing:** Sonal, Alex E. Yuan, Xueqin Yang, Wenying Shou.

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
