## [Editor Report · Decision Letter 0]

18 May 2023

Dear Sonal, 

Thank you for submitting your manuscript entitled "When is an auxotroph not an auxotroph: how budding yeast lacking MET17 collectively overcome their metabolic defect" for consideration as a Research Article by PLOS Biology (under the terms of our "anti-scooping" policy).

Your manuscript has now been evaluated by the PLOS Biology editorial staff, as well as by an academic editor with relevant expertise, and I'm writing to let you know that we would like to send your submission out for external peer review.

Once your full submission is complete, your paper will undergo a series of checks in preparation for peer review. After your manuscript has passed the checks it will be sent out for review. To provide the metadata for your submission, please Login to Editorial Manager (https://www.editorialmanager.com/pbiology) within two working days, i.e. by May 22 2023 11:59PM.

Kind regards,

Roli

Roland Roberts, PhD

Senior Editor

PLOS Biology

rroberts@plos.org

---

## [Decision Letter · Decision Letter 1]

28 Jun 2023

Dear Dr Sonal,

My name is Luke Smith - I am an editor at PLOS Biology, and am handling your manuscript "When is an auxotroph not an auxotroph: how budding yeast lacking MET17 collectively overcome their metabolic defect" on behalf of my colleague, Roli, who is out of the office this week. Thank you for your patience while your manuscript was peer reviewed at PLOS Biology. Your study has now been evaluated by the PLOS Biology editors, an Academic Editor with relevant expertise, and by several independent reviewers. In light of the reviews, which you will find at the end of this email, we would like to invite you to revise the work to thoroughly address the reviewers' reports.

As you will see below, while the reviewers are fairly positive about the study overall, they raised a number of important points for you to consider in the revision. We would strongly encourage you to address the reviewer comments with new experimental data, where appropriate, including by adding the controls asked for by Reviewer 2 and the growth experiment asked for by Reviewer 1 - as we think these analyses would strengthen the study.

Given the extent of revision needed, we cannot make a decision about publication until we have seen the revised manuscript and your response to the reviewers' comments. Your revised manuscript is likely to be sent for further evaluation by all or a subset of the reviewers.

**IMPORTANT - SUBMITTING YOUR REVISION**

*Re-submission Checklist*

*Published Peer Review*

*PLOS Data Policy*

*Blot and Gel Data Policy*

Sincerely,

Luke Smith, PhD

Senior Editor 

PLOS Biology

lsmith@plos.org

on behalf of

Roland Roberts, PhD

Senior Editor

PLOS Biology

rroberts@plos.org

REVIEWS:

Reviewer #1: The paper by Sonal et al. adds to the recent discovery that yeast cells lacking MET17 can grow on sulfate when the cell number is high. At those conditions, the buildup of hydrogen sulfide in met17 mutants is used by Hsu1, and sulfur is assimilated. Overall, the authors do a good job of describing these mutants' growth in various conditions and conclude that the efficiency of using the accumulated hydrogen sulfide is low, and the reaction happens only at high cell densities. The results reinforce each other in the context of the other recently published papers on the same topic. 

Some points that need to be considered:

- Overall, the central unanswered question now is the putative function of Hsu1 in wild-type cells since the authors find no fitness advantage that HSU1 confers to wild-type yeast cells (not met17 mutants) during sulfur starvation. They speculate on its role during some sulfur-related stress responses. This should be tested experimentally. Upon exposure to cadmium, which triggers a well-described increase in glutathione synthesis, what is the viability of cells lacking or over-expressing HSU1? Additional stress conditions they mention (e.g., heat) could also be tested. Providing these answers is significant since Yu et al. reported some fitness effects attributed to Hsu1, which were not seen by Sonal et al. 

- Can over-expression of Hsu1 shorten the lag in the growth of met17 mutants? In the model, they propose that Hsu1 is not efficient and works only at high sulfide concentrations. One might expect that providing more Hsu1 would increase the reaction rate, in a largely linear manner.

Reviewer #2: In "When is an auxotroph not an auxotroph: how budding yeast lacking MET17 collectively overcome their metabolic defect" Sonal et al. describe and try to explain a leaky auxotrophy in met17_delta yeast strains. There is novelty and value to this manuscript, especially for the functional study of the HSU1 ORF and for the wine industry. However, there are some major issues that I would like to see addressed before recommending it for publication. Below are my main criticisms.

Throughout the study the growth behaviour of various strains from deletion collections are studied. However, the omission of positive and negative controls (such as a prototrophic and a "true auxotrophic" strain, respectively) in each experiment makes it impossible to assess the effects of each perturbation. Examples include Figures 2, 3 and 5. This is a critical point for me.

It is well known that carbon dioxide can induce growth in yeast cells, yet this possible explanation for the growth behaviour of cultures sharing headspace is never addressed. 

The authors note a background strain-dependent difference in met17_delta phenotype. I would appreciate a discussion of why this could be, especially since RM11 is a wine isolate and the importance of sulfide content on wine quality. 

The manuscript would benefit from a clearer integration of the proposed mathematical model and the observed phenotypes. Suggestions include validating the model's predictive capabilities through new experiments and/or using the model to explain the mechanisms behind the phenotypes.

I would appreciate a clearer distinction between the Results and Discussion sections. For example, L230-240 and L435-466 within Results would be better fit for Discussion.

The manuscript would benefit from proofreading, especially figure captions and references, strain number corresponding to genotype and background, and Table S1, incuding the exact parent strain(s) of the constructed as well as clear references.

---

## [Editor Report · Decision Letter 2]

13 Nov 2023

Dear Sonal,

Thank you for your patience while we considered your revised manuscript "When is an auxotroph not an auxotroph: how budding yeast lacking MET17 collectively overcome their metabolic defect" for publication as a Research Article at PLOS Biology. This revised version of your manuscript has been evaluated by the PLOS Biology editors and the Academic Editor.

Based on our Academic Editor's assessment of your revision, we are likely to accept this manuscript for publication, provided you satisfactorily address the following data and other policy-related requests:

IMPORTANT: It would be great if we could publish this paper this year, given the prior related publications. Please could you turn these requests around swiftly, and we'll try our best expedite this end of the process! Please address the following:

a) The Academic Editor said: One extremely minor comment: in line 227 they say "sodium hydroxide" but they mean "sodium hydrosulfide" (NaHS).

b) Please could you make your title more accurately reflect your findings? We suggest something like "Budding yeast lacking MET17 collectively overcome their metabolic defect by density-dependent accumulation of hydrogen sulfide gas"

c) Please address my Data Policy requests below; specifically, we need you to supply the numerical values underlying Figs 1CD, 2ABCDEFG, 3CE, 4E, 5CEF, S1AB, S2AB, S3AB, S5AB, S6ABC, S7ABC, S8, S9ABCD, S10, S11, S12ABCD, S13ABC, S14AB, S15, S16, S17B, S18, either as a supplementary data file or as a permanent DOI’d deposition. I note that you already have an associated GitHub deposition (https://github.com/aeyuan/H2S-mediated-density-dependence), but this only claims to contain the means to generate Figs ST-1 to ST-4.

d) Because Github depositions can be readily changed or deleted, please make a permanent DOI’d copy (e.g. in Zenodo) and provide this URL (see below).

e) Please cite the location of the data clearly in all relevant main and supplementary Figure legends, e.g. “The data underlying this Figure can be found in S1 Data” or “The data underlying this Figure can be found in https://doi.org/10.5281/zenodo.XXXXX”

f) Please make any other custom code available, either as a supplementary file or as part of your Zenodo deposition.

We expect to receive your revised manuscript within one week. 

*Published Peer Review History*

*Press*

Sincerely,

Roli

Roland Roberts, PhD

Senior Editor,

rroberts@plos.org,

PLOS Biology

DATA POLICY:

Regardless of the method selected, please ensure that you provide the individual numerical values that underlie the summary data displayed in the following figure panels as they are essential for readers to assess your analysis and to reproduce it: Figs 1CD, 2ABCDEFG, 3CE, 4E, 5CEF, S1AB, S2AB, S3AB, S5AB, S6ABC, S7ABC, S8, S9ABCD, S10, S11, S12ABCD, S13ABC, S14AB, S15, S16, S17B, S18. NOTE: the numerical data provided should include all replicates AND the way in which the plotted mean and errors were derived (it should not present only the mean/average values).

CODE POLICY

Per journal policy, as the code that you have generated is important to support the conclusions of your manuscript, we require that you make it available without restrictions upon publication. Please ensure that the code is sufficiently well documented and reusable, and that your Data Statement in the Editorial Manager submission system accurately describes where your code can be found.

SPECIES INDICATED IN THE ABSTRACT? 

- Please note that per journal policy, the model system/species studied should be clearly stated in the abstract of your manuscript. 

We require the original, uncropped and minimally adjusted images supporting all blot and gel results reported in an article's figures or Supporting Information files. We will require these files before a manuscript can be accepted so please prepare and upload them now. Please carefully read our guidelines for how to prepare and upload this data: https://journals.plos.org/plosbiology/s/figures#loc-blot-and-gel-reporting-requirements

DATA NOT SHOWN?

---

## [Editor Report · Decision Letter 3]

20 Nov 2023

Dear Sonal,

Thank you for the submission of your revised Research Article "Collective production of hydrogen sulfide gas enables budding yeast lacking MET17 to overcome their metabolic defect" for publication in PLOS Biology. On behalf of my colleagues and the Academic Editor, Mark Siegal, I'm pleased to say that we can in principle accept your manuscript for publication, provided you address any remaining formatting and reporting issues. These will be detailed in an email you should receive within 2-3 business days from our colleagues in the journal operations team; no action is required from you until then. Please note that we will not be able to formally accept your manuscript and schedule it for publication until you have completed any requested changes.

PRESS :We frequently collaborate with press offices. If your institution or institutions have a press office, please notify them about your upcoming paper at this point, to enable them to help maximise its impact. If the press office is planning to promote your findings, we would be grateful if they could coordinate with biologypress@plos.org. If you have previously opted in to the early version process, we ask that you notify us immediately of any press plans so that we may opt out on your behalf.

Sincerely,

Roli

Senior Editor

PLOS Biology

rroberts@plos.org